# Solving the Granularity Mismatch: Hierarchical Preference Learning for Long-Horizon LLM Agents

**Heyang Gao**[1,2,*], **Zexu Sun**[3,*], **Erxue Min**[3], **Hengyi Cai**[3],
**Shuaiqiang Wang**[3], **Dawei Yin**[3], **Xu Chen**[1,2,†]

[1] Guangdong Laboratory of Artificial Intelligence and Digital Economy (SZ)

[2] Gaoling School of Artificial Intelligence, Renmin University of China

[3] Baidu Inc.

{gaoheyang,xu.chen}@ruc.edu.cn, sunzexu0826@gmail.com

## Abstract

Large Language Models (LLMs) as autonomous agents are increasingly tasked with solving complex, long-horizon problems. Aligning these agents via preference-based methods like Direct Preference Optimization (DPO) is a promising direction, yet it faces a critical granularity mismatch. Trajectory-level DPO provides stable signals but blur where credit should be assigned within long trajectories, whereas step-level DPO offers fine-grained supervision but can be statistically noisy and data-inefficient when Monte Carlo rollouts are limited, and can be hard to fully exploit multi-step structured behaviors that only reveal their effect over several actions. To balance this trade-off, we introduce **H**ierarchical **P**reference **L**earning (HPL), a hierarchical framework that optimizes LLM agents by leveraging preference signals at multiple, complementary granularities. While HPL incorporates trajectory- and step-level DPO for global and local policy stability, its core innovation lies in group-level preference optimization guided by a dual-layer curriculum. HPL first decomposes expert trajectories into semantically coherent action groups and then generates contrasting suboptimal groups to enable preference learning at a fine-grained, sub-task level. Then, instead of treating all preference pairs equally, HPL introduces a curriculum scheduler that organizes the learning process from simple to complex. This curriculum is structured along two axes: the group length, representing sub-task complexity, and the sample difficulty, defined by the reward gap between preferred and dispreferred action groups. Experiments on three challenging agent benchmarks show that HPL outperforms existing state-of-the-art methods. Our analyses demonstrate that the hierarchical DPO loss effectively integrates preference signals across multiple granularities, while the dual-layer curriculum is crucial for enabling the agent to solve a wide range of tasks, from simple behaviors to complex multi-step sequences.

## 1 Introduction

Large Language Models (LLMs) have evolved from static question-answering systems into autonomous agents capable of perceiving, reasoning, and acting within complex, open-ended environments (Li et al., 2024; Gou et al., 2025). This transformation has powered a new generation of applications, from embodied assistants that navigate simulated homes (Shridhar et al., 2021) to web navigators that execute multi-step online tasks (Zheng et al., 2024; Furuta et al., 2024). Unlike single-turn tasks, these agent-environment interactions unfold in multi-turn loops over extended periods (Wang et al., 2024a). This paradigm shift introduces a core challenge: long-horizon planning and decision-making, where the agent must execute a coherent sequence of actions to succeed.

---

* Equal contribution.

† Corresponding author.

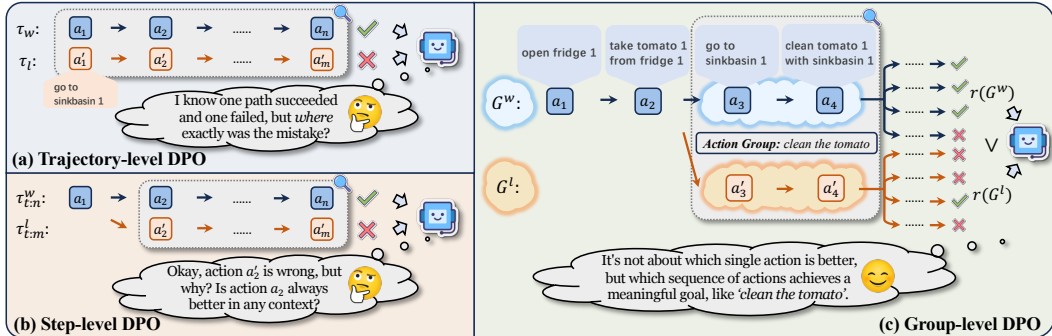

Figure 1: Conceptual comparison of different DPO granularities. While **(a)** trajectory-level DPO provides a coarse but stable signal and **(b)** step-level DPO offers focused but potentially noisy supervision, **(c)** our proposed Group-level DPO learns from semantically coherent action groups, which provides a structured signal, enabling the agent to reason at the sub-task level.

To equip agents for such tasks, Reinforcement Learning (RL) has become a crucial recipe for post-training LLMs (Liu et al., 2024). Online RL methods like PPO (Sutton et al., 1998; Schulman et al., 2017) often entail substantial computational costs, high sample inefficiency, and risky, inefficient exploration in vast action spaces. These challenges have motivated approaches that reduce reliance on online interaction by learning from static datasets collected in advance. Direct Preference Optimization (DPO) (Rafailov et al., 2023) directly aligns agent policies using preference pairs (*e.g.*, expert vs. suboptimal behaviors) without requiring costly environment interaction or an explicitly trained reward model.

However, applying DPO to long-horizon agent tasks reveals a fundamental challenge we term the **granularity mismatch**. On one hand, trajectory-level DPO, such as ETO (Song et al., 2024), compares entire trajectories and yields stable, low-variance feedback aligned with final outcomes, but it provides limited resolution for credit assignment, which is hard to tell which segment of a long interaction actually determined success or failure. On the other hand, step-level DPO, employed by methods such as IPR (Xiong et al., 2024), attributes preferences to individual decisions by estimating the expected return from each decision point via Monte Carlo rollouts. However, this fine-grained focus poses practical challenges in the finite-data, limited-rollout regime. Supervision becomes fragmented across many decision points, so each step is updated from only a few noisy rollouts, and it can be hard to fully exploit multi-step structured behaviors whose contribution to success only becomes apparent when several actions are considered jointly. For instance, the sub-task of "retrieving an apple from the fridge" is composed of a chain of actions—navigating, opening, and taking—whose collective value cannot be captured by rewarding any single action in isolation.

To resolve this dilemma, we introduce **H**ierarchical **P**reference **L**earning (HPL), a hierarchical framework that optimizes LLM agents by leveraging preference signals at multiple, synergistic granularities. HPL first addresses the granularity mismatch by incorporating DPO losses at the trajectory, action, and the action-group levels. The group-level view provides both a **structural prior** by focusing supervision on sub-trajectories that are more likely to encode reusable skills and a **statistical benefit** by aggregating the contribution of multiple actions into one decision unit, which reduces variance relative to per-step Monte Carlo estimates under a fixed rollout budget. Beyond merely combining these losses, the core innovation of HPL is a dual-layer curriculum learning strategy that guides the training process. This curriculum systematically organizes the learning path from simple to complex along two orthogonal axes: sub-task complexity, defined by the length of an action group, and sample difficulty, measured by the reward gap between preferred and dispreferred behaviors. By first mastering simple, easily distinguishable sub-tasks, the agent builds a foundation before progressing to more complex challenges.

Our main contributions are summarized as follows:

- We identify and address the granularity mismatch problem in preference-based agent alignment by proposing a novel hierarchical framework that integrates preference signals at three distinct levels: the coarse trajectory-level, the fine-grained step-level, and a intermediate action-group level.

- We introduce HPL, a novel training paradigm with a dual-layer curriculum learning strategy. This is the first work to apply a structured curriculum to action-group level preference optimization, dynamically scheduling samples based on both task complexity and distinguishability.

- We design and systematically evaluate a range of action grouping strategies, from simple heuristics to a semantic-based approach, to generate meaningful sub-tasks for our framework.

- We demonstrate through extensive experiments on three diverse and challenging long-horizon benchmarks that HPL significantly outperforms existing state-of-the-art methods, establishing a more effective and principled paradigm for preference-based training of LLM agents from fixed datasets after a one-shot exploration phase.

## 2 RELATED WORK

**LLM-based Agents.** The remarkable reasoning and instruction-following capabilities of modern LLMs have enabled their use as autonomous agents capable of tackling a diverse array of complex, interactive tasks (Li et al., 2024; Gou et al., 2025; Tang et al., 2025; Wang et al., 2025b). Initial approaches primarily leveraged the in-context learning abilities of LLMs through prompts like Re-Act (Yao et al., 2023) and Reflexion (Shinn et al., 2023) to elicit multi-step reasoning and action generation. This focus on inference-time computation has recently evolved into the broader paradigm of test-time scaling (TTS), where allocating additional compute during generation (Zhang et al., 2025b). However, to enhance the performance of open-source models beyond zero-shot prompting and and test-time strategies, a subsequent line of work has focused on fine-tuning agents on collected trajectory data (Chen et al., 2023). These methods range from standard Supervised Fine-Tuning (SFT) on expert demonstrations (Zeng et al., 2023; Chen et al., 2024) to more advanced techniques that learn from preference data, such as contrasting successful and failed trajectories to optimize for final outcomes (Song et al., 2024). While effective, these fine-tuning paradigms often treat entire trajectories as monolithic data points (Wang et al., 2024b), overlooking the fine-grained procedural knowledge embedded within the interaction trajectory.

**Process Supervision.** To address the limitations of outcome-based rewards, particularly the challenge of credit assignment in long-horizon tasks, a growing body of research has explored process supervision (Luo et al., 2024; Xiong et al., 2025). The core idea is to provide agents with more granular feedback at intermediate steps of a task. Early efforts in this area often relied on costly human annotations to label the correctness of each step (Lightman et al., 2023). To automate this, recent methods have proposed various techniques to estimate step-level rewards, such as using Monte Carlo rollouts to predict the future outcome from an intermediate state (Xiong et al., 2024) or training a separate reward model to predict the value of each action (Choudhury, 2025; Wang et al., 2025a). These approaches typically use the estimated step-level rewards to guide the agent via reinforcement learning (Feng et al., 2025; Zhang et al., 2025a) or DPO at the single-action level.

## 3 METHODOLOGY

In this section, we present our novel agent alignment framework **H**ierarchical **P**reference **L**earning (HPL) as depicted in Figure 2. We detail the principal phases of our method below: initial policy bootstrapping through behavior cloning, the generation of multi-granularity preference data, the design of our dual-layer curriculum scheduler, and finally, the hierarchical optimization objective.

### 3.1 PROBLEM SETTING

In this work, HPL follow the same two-stage protocol as previous work (Song et al., 2024; Xiong et al., 2024) that combines a one-shot exploration phase with purely offline preference optimization.

- **Fixed exploration and labeling.** A frozen reference policy $\pi_{\text{ref}}$ interacts with the environment once to collect a pool of interaction traces, including both full trajectories and partial segments. We then run Monte Carlo rollouts with $\pi_{\text{ref}}$ on this static pool to derive different level reward estimates, constructing the preference datasets. Importantly, the policy $\pi_\theta$ is never updated during this phase, and no further data collection is performed once the datasets are built.

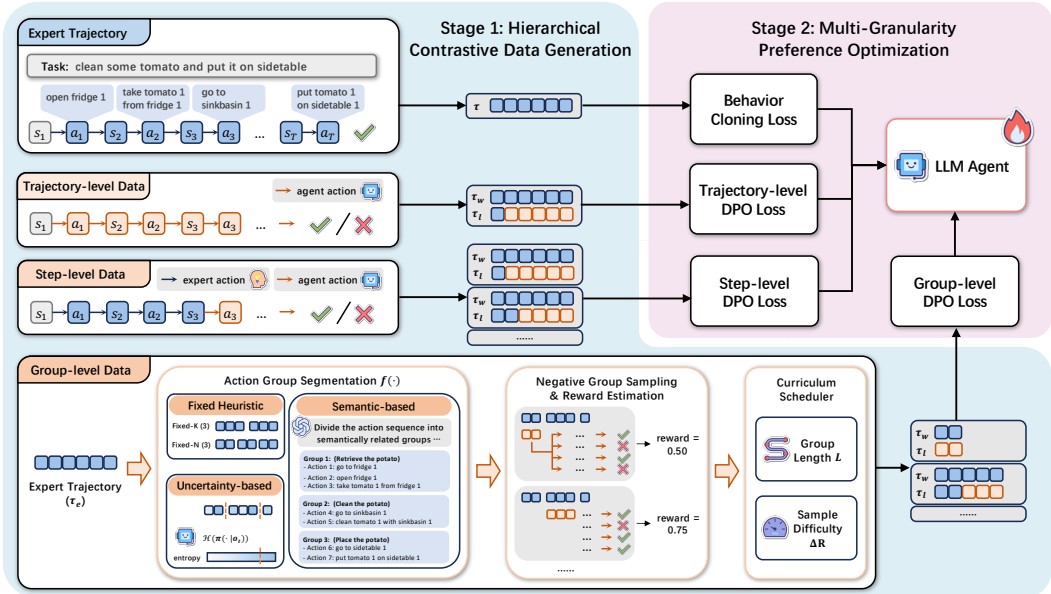

Figure 2: An overview of our proposed framework, HPL. Stage 1 generates hierarchical preference data with Action Group Segmentation component. Stage 2 then optimizes the agent with a composite objective, where the training is guided by dual-layer curriculum scheduler.

- **Offline preference optimization.** Given these fixed datasets, preference-based methods train the target policy $\pi_\theta$ using DPO-style objectives at different levels (trajectory, step, and group), without any additional environment interaction or reward queries.

We therefore view our setting as offline preference optimization after a one-shot exploration phase, which is distinct from fully online RL methods such as PPO (Schulman et al., 2017) that continuously interleave policy updates with new environment rollouts throughout training.

## 3.2 BOOTSTRAPPING VIA EXPERT BEHAVIOR CLONING

To equip the base model with fundamental task-solving capabilities, we perform behavior cloning on a dataset of expert trajectories $\mathcal{D}_{\text{expert}} = \{(u, \tau^*)^{(i)}\}_{i=1}^{|\mathcal{D}_{\text{expert}}|}$, where $u$ is the task instruction and $\tau^*$ is the corresponding expert trajectory. Each trajectory $\tau$ is a sequence of alternating states and actions $\tau = (s_1, a_1, s_2, a_2, \ldots, s_T, a_T)$. A state $s_t$ is a textual description of the environment, and an action $a_t$ is a textual command generated by the agent. This process aims to maximize the likelihood of the expert's actions. The loss is defined as:

$$\mathcal{L}_{\text{BC}}(\theta; \mathcal{D}_{\text{expert}}) = -\mathbb{E}_{(u, \tau^*) \sim \mathcal{D}_{\text{expert}}} \left[ \sum_{t=1}^{|\tau^*|} \log \pi_\theta(a_t^* | s_t^*, u, \tau_{<t}^*) \right], \tag{1}$$

where $\tau_{<t}^*$ represents the history $(s_1^*, a_1^*, \ldots, s_{t-1}^*, a_{t-1}^*)$. This initial cloning step yields a competent base agent, which serves as our reference policy $\pi_{\text{ref}}$ for the subsequent optimization stages.

## 3.3 HIERARCHICAL CONTRASTIVE DATA GENERATION

After obtaining a competent reference policy $\pi_{\text{ref}}$ via behavior cloning (Section 3.2), the next stage is to generate a rich, multi-layered dataset for preference optimization. This is achieved by having $\pi_{\text{ref}}$ interact with the environment to produce a diverse set of suboptimal trajectories. By contrasting these with expert trajectories, we construct three distinct preference datasets at the trajectory, action, and group granularities, as illustrated in the left panel of Figure 2.

### 3.3.1 DATA GENERATION AT TRAJECTORY AND ACTION LEVELS

**Trajectory-Level Data.** This dataset provides a coarse, outcome-based learning signal. For each expert trajectory $\tau_w$ from $\mathcal{D}_{\text{expert}}$, we use $\pi_{\text{ref}}$ to generate a corresponding full trajectory $\tau_l$. If the outcome reward of $\tau_l$ is lower than that of $\tau_w$, we form a preference pair $(\tau_w, \tau_l)$. This process yields a trajectory-level dataset $\mathcal{D}_{\text{traj}} = \{(u, \tau_w, \tau_l)^{(i)}\}$, where $u$ is the task instruction.

**Step-Level Data.** To provide a finer-grained, process-oriented signal, we adopt the methodology from IPR (Xiong et al., 2024). At each step $t$ of an expert trajectory, we use the history $\tau_{<t}$ as a prompt for our reference agent $\pi_{\text{ref}}$ to generate an alternative action $\hat{a}_t$ and complete the rest of the trajectory, yielding $\tau_{t:m}^l$. This is contrasted with the expert's subsequent trajectory $\tau_{t:n}^w$. This creates a preference pair conditioned on the shared history, resulting in a step-level preference dataset $\mathcal{D}_{\text{step}} = \{(\tau_{<t}, \tau_{t:n}^w, \tau_{t:m}^l)^{(i)}\}$.

### 3.3.2 GROUP-LEVEL DATA GENERATION VIA ACTION GROUP SEGMENTATION

To bridge the gap between coarse trajectories and single actions, we introduce the core concept of an **action group**. These groups serve as an intermediate unit of reasoning, ideally corresponding to semantically coherent sub-tasks for more effective credit assignment. The generation of group-level data involves two key steps: segmenting trajectories into groups and then estimating a quantitative reward for each group.

First, we apply a segmentation function $f(\cdot)$ to partition expert trajectory $\tau_w$ into corresponding action groups $\{G_{w,i}\}_{i=1}^N$. We design and investigate four distinct segmentation strategies:

**Fixed Heuristic Strategies.** As baselines, we consider two straightforward methods based on length. *Fixed-N Groups* divides a trajectory into a fixed number $N$ of equal-length groups. *Fixed-K Size* creates groups with $K$ consecutive action steps each. While simple, these methods are agnostic to the task's semantic structure and risk making arbitrary cuts.

**Uncertainty-Based Segmentation.** This adaptive strategy is based on the intuition that a policy's uncertainty often increases at sub-task boundaries (Guo et al., 2025). We leverage the entropy of the reference policy's action distribution, $H(\pi_{\text{ref}}(\cdot|o_t))$, as a proxy for uncertainty. A boundary is inserted after action $a_{t-1}$ if the entropy at step $t$ exceeds a predefined threshold $\epsilon$.

**Semantic Segmentation.** To achieve the most meaningful partitions, we employ a powerful, pretrained LLM (*e.g.*, GPT-4o) as an off-the-shelf "semantic segmenter". We provide the full text transcript of a trajectory to the model and prompt it to partition the sequence into high-level subtasks based on their apparent goals (*e.g.*, "find an object", "operate an appliance"). This method is expected to yield the highest quality segmentations.

Once an expert trajectory is partitioned into a sequence of winning groups $\{G_{w,i}\}$, we construct the preference pairs required for our group-level optimization. For each expert action group $G_{w,i}$, which begins from a context $c_i$ (*i.e.*, the history of all preceding steps), we generate a corresponding losing group $G_{l,i}$. This is achieved by sampling a new action sequence of the same length from the reference policy $\pi_{\text{ref}}(\cdot|c_i)$. This length-constrained sampling ensures a fair, apples-to-apples comparison between the expert and suboptimal behaviors. The resulting tuples $(c_i, G_{w,i}, G_{l,i})$ form a rich dataset of fine-grained preference pairs, which are the fundamental training units for HPL.

### 3.3.3 GROUP-LEVEL REWARD ESTIMATION

After segmenting trajectories into groups, we need a quantitative reward estimate for each group to filter data and enable our curriculum learning strategy (detailed in Section 3.4). We define the reward of an action group $G_i$, which ends at timestep $t_i$ with history $\tau_{<t_i}$, as the expected final outcome reward of trajectories completed from that point. Given the difficulty of direct computation, we estimate this value using Monte Carlo (MC) sampling. Specifically, we use our reference policy $\pi_{\text{ref}}$ to perform $M$ stochastic rollouts starting from the state after $G_i$ has been executed. The estimated reward for the action group $G_i$, denoted $\hat{r}(G_i)$, is the average of the final outcome rewards $R(\cdot)$ from these rollouts:

$$\hat{r}(G_i) = \frac{1}{M} \sum_{j=1}^{M} R(\tau_i^{(j)}), \quad \text{where } \{\tau_i^{(j)}\}_{j=1}^M = \text{MC}^{\pi_{\text{ref}}}(\tau_{<t_i}; M). \tag{2}$$

This MC estimation is applied to every winning group $G_{w,i}$ and losing group $G_{l,i}$ to obtain their rewards. With these components, we finalize our group-level dataset $\mathcal{D}_{\text{group}} = \{(c, G_w, G_l)^{(i)}\}$, where each entry contains a context, a preference pair of groups, and their estimated rewards.

While step-level MC estimation provides a principled, localized estimate of the expected return at each decision point, in practice the rollout budget per state is small. Under this limited-rollout regime, per-step estimates can have high variance, which makes purely step-level preference learning statistically inefficient. In contrast, our group-level rewards aggregate the outcomes of multiple actions into a single supervision unit, amortizing the same rollout budget over longer sub-trajectories and better capturing their joint contribution to task success.

## 3.4 DUAL-LAYER CURRICULUM LEARNING

Statically mixing the group-level preference data from all difficulties for training can be suboptimal, as it may expose the model to highly complex samples before it has developed foundational skills, leading to unstable or inefficient learning. To address this, we introduce the core innovation of our framework: a **dual-layer curriculum learning** strategy. This strategy dynamically organizes the training process to mimic an efficient human learning path, from simple concepts to complex ones.

**The 2D Curriculum Matrix.** As illustrated in Figure 3, our curriculum is conceptualized as a two-dimensional difficulty matrix. We categorize each group-level preference pair $(G_w, G_l)$ along two orthogonal axes:

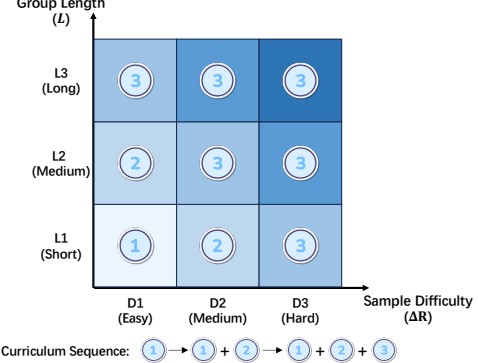

- **Sub-task Complexity (Y-axis):** This dimension is measured by the **Group Length** ($L$). Shorter action groups (*e.g.*, 1-3 steps) correspond to simple, fundamental skills, while longer groups represent more complex, multi-step behaviors that require longer-term planning.

- **Sample Discriminability (X-axis):** This dimension is measured by the **Sample Difficulty** ($\Delta R$), defined as the difference between the estimated rewards of the winning and losing groups: $\Delta R = \hat{r}(G_w) - \hat{r}(G_l)$. A large $\Delta R$ indicates an easy-to-distinguish sample where the losing group is clearly inferior. A small $\Delta R$ represents a hard-to-distinguish sample that requires finer judgment from the agent for successful policy refinement.

Figure 3: Illustration of the dual-layer curriculum scheduler with group length ($L$) and sample difficulty ($\Delta R$). The training follows a three-phase schedule.

Based on these two axes, we partition the group-level dataset $\mathcal{D}_{\text{group}}$ into a 3x3 grid of data buckets, denoted as $\mathcal{B}_{L,D}$, where $L, D \in \{1, 2, 3\}$ represent the levels of length and difficulty, respectively.

**The Curriculum Schedule.** Our training process is not a single pass over the mixed data, but a staged schedule that progressively expands the training set, guiding the model along a path of increasing difficulty. The schedule consists of three distinct phases:

1. **Phase 1 (Foundational Skills):** Initially, the model is trained exclusively on the easiest data bucket, $\mathcal{B}_{1,1}$ (short length, easy difficulty). This allows the agent to quickly and stably learn the most fundamental and unambiguous skills without being distracted by more complex scenarios.

2. **Phase 2 (Expanding Complexity):** After the initial phase, we expand the training data to include $\mathcal{B}_{1,1} \cup \mathcal{B}_{1,2} \cup \mathcal{B}_{2,1}$. In this stage, the agent begins to tackle harder (less distinguishable) short-horizon tasks while also being introduced to simple medium-horizon skills, effectively broadening its capabilities.

3. **Phase 3 (Full-Scale Tuning):** Finally, the training set is expanded to include all nine buckets ($\bigcup_{L,D} \mathcal{B}_{L,D}$). The agent now fine-tunes its policy on the full spectrum of complexities and difficulties, mastering the most challenging and nuanced aspects of the tasks.

This staged exposure ensures a smooth learning gradient, building agent's expertise from the ground up and preventing it from being overwhelmed by difficult samples early in the training process.

### 3.5 MULTI-GRANULARITY PREFERENCE OPTIMIZATION

In the final stage, we optimize the policy $\pi_\theta$ using a composite loss function that integrates signals from all three granularities. This approach ensures that the agent not only learns from high-level outcomes and fine-grained sub-tasks but also stays grounded in the expert's behavior. The final loss includes a sum of three components:

**Trajectory-Level DPO Loss ($\mathcal{L}_{\text{traj-DPO}}$).** To learn from the overall outcome, we apply a DPO loss on the trajectory-level dataset $\mathcal{D}_{\text{traj}}$. This loss encourages the policy to assign a higher likelihood to the entire successful trajectory over the failed one:

$$\mathcal{L}_{\text{traj-DPO}}(\theta; \mathcal{D}_{\text{traj}}) = -\mathbb{E}_{(\tau_w, \tau_l) \sim \mathcal{D}_{\text{traj}}} \left[ \log \sigma \left( \beta \log \frac{\pi_\theta(\tau_w|u)}{\pi_{\text{ref}}(\tau_w|u)} - \beta \log \frac{\pi_\theta(\tau_l|u)}{\pi_{\text{ref}}(\tau_l|u)} \right) \right]. \quad (3)$$

**Step-Level DPO Loss ($\mathcal{L}_{\text{step-DPO}}$).** Drawing from Xiong et al. (2024), this loss uses $\mathcal{D}_{\text{step}}$ to provide step-level supervision by comparing the entire future from a decision point:

$$\mathcal{L}_{\text{step-DPO}}(\theta; \mathcal{D}_{\text{step}}) = -\mathbb{E}_{(\tau_{<t}, \tau_{t:n}^w, \tau_{t:m}^l) \sim \mathcal{D}_{\text{step}}} \left[ \log \sigma \left( \beta \log \frac{\pi_\theta(\tau_{t:n}^w|\tau_{<t})}{\pi_{\text{ref}}(\tau_{t:n}^w|\tau_{<t})} - \beta \log \frac{\pi_\theta(\tau_{t:m}^l|\tau_{<t})}{\pi_{\text{ref}}(\tau_{t:m}^l|\tau_{<t})} \right) \right]. \quad (4)$$

**Group-Level DPO Loss ($\mathcal{L}_{\text{group-DPO}}$).** This is the core component of our framework, providing mid-level supervision. We apply the DPO loss to the group-level dataset $\mathcal{D}_{\text{group}}$, comparing corresponding action groups:

$$\mathcal{L}_{\text{group-DPO}}(\theta; \mathcal{D}_{\text{group}}) = -\mathbb{E}_{(c, G_w, G_l) \sim \mathcal{D}_{\text{group}}} \left[ \log \sigma \left( \beta \log \frac{\pi_\theta(G_w|c)}{\pi_{\text{ref}}(G_w|c)} - \beta \log \frac{\pi_\theta(G_l|c)}{\pi_{\text{ref}}(G_l|c)} \right) \right]. \quad (5)$$

We briefly analyze the bias-variance properties of this group-level objective in the following proposition, with a full derivation provided in Appendix G.

**Proposition 1** (Bias-variance trade-off of group-level DPO loss). *Let $T$ denote the trajectory length, $\gamma \in [0, 1)$ the discount factor, and $R_{max}$ the maximum reward. Let $\mathcal{L}_{traj}$, $\mathcal{L}_{step}$, and $\mathcal{L}_{group}(k)$ denote the empirical losses of trajectory-level, step-level, and group-level DPO with group length $k < T$, respectively. Then there exists a constant $C > 0$ depending only on $(\gamma, \pi_{ref})$ such that for every $\epsilon \in (0, 1)$ the choice $k(\epsilon) = \left\lceil \log_\gamma \left( \frac{(1-\gamma)\epsilon}{2\beta R_{max}} \right) \right\rceil$ satisfies*

$$\text{Bias}(\mathcal{L}_{group}(k)) \leq \min\{\text{Bias}(\mathcal{L}_{traj}), \text{Bias}(\mathcal{L}_{step})\} + \epsilon, \quad (6)$$

$$\text{Var}(\mathcal{L}_{group}(k)) \leq \frac{C \log(1/\epsilon)}{T} \min\{\text{Var}(\mathcal{L}_{traj}), \text{Var}(\mathcal{L}_{step})\}. \quad (7)$$

Hence, by setting $k = \Theta(\log(1/\epsilon))$, group-level DPO loss simultaneously improves the variance by a factor $\Omega(T/\log(1/\epsilon))$ while incurring at most an additive bias of $\epsilon$ over the other two losses.

The final training objective combines these losses:

$$\mathcal{L}_{final}^{(s)} = \mathcal{L}_{\text{BC}} + \mathcal{L}_{\text{traj-DPO}} + \mathcal{L}_{\text{step-DPO}} + \mathcal{L}_{\text{group-DPO}}^{(s)}, \quad (8)$$

where the group-level loss $\mathcal{L}_{\text{group-DPO}}^{(s)}$ for curriculum stage s is computed over a dynamically selected subset of data, $\mathcal{D}_{\text{group}}^{(s)}$, which is determined by our curriculum scheduler (Section 3.4). The data subset $\mathcal{D}_{\text{group}}^{(s)}$ for each stage is constructed from the 2D curriculum matrix buckets $\mathcal{B}_{L,D}$ as follows:

$$\mathcal{D}_{\text{group}}^{(s)} = \begin{cases} \mathcal{B}_{1,1} & \text{if } s = 1 \text{ (Phase 1)} \\ \mathcal{B}_{1,1} \cup \mathcal{B}_{1,2} \cup \mathcal{B}_{2,1} & \text{if } s = 2 \text{ (Phase 2)} \\ \bigcup_{L,D} \mathcal{B}_{L,D} & \text{if } s = 3 \text{ (Phase 3)} \end{cases} \quad (9)$$

## 4 EXPERIMENTS

In this section, we conduct a series of experiments to comprehensively evaluate the performance of our HPL framework. Our evaluation is designed to answer the following key research questions:

Table 1: Performance comparison of HPL and baselines across agent benchmarks over 3 random seeds. All methods are evaluated using Qwen2.5-1.5B-Instruct and Qwen2.5-7B-Instruct as base models. The best and second-best results are highlighted in **bold** and with an underline, respectively.

| Models | ALFWorld | | WebShop | | InterCode-SQL | | Average |
|---|---|---|---|---|---|---|---|
| | seen | unseen | avg. reward | success rate | avg. reward | success rate | |
| GPT-4o | 36.43 | 32.09 | 55.26 | 18.50 | 28.50 | 28.50 | 33.21 |
| Gemini-2.5-Pro | 55.71 | 49.25 | 49.56 | 19.50 | 68.42 | 66.00 | 51.40 |
| **Qwen2.5-1.5B-Instruct** | 2.14 | 0.00 | 36.09 | 10.50 | 5.50 | 5.50 | 9.95 |
| SFT | $60.95_{\pm1.09}$ | $57.96_{\pm1.88}$ | $56.56_{\pm0.69}$ | $26.00_{\pm0.50}$ | $56.24_{\pm0.61}$ | $54.33_{\pm0.76}$ | $52.01_{\pm0.43}$ |
| RFT (Yuan et al., 2023) | $61.19_{\pm1.80}$ | $60.95_{\pm0.86}$ | $57.66_{\pm1.45}$ | $28.17_{\pm1.04}$ | $58.08_{\pm0.64}$ | $56.67_{\pm0.29}$ | $53.79_{\pm0.40}$ |
| ETO (Song et al., 2024) | $65.48_{\pm3.60}$ | $66.42_{\pm2.24}$ | $56.57_{\pm0.22}$ | $28.00_{\pm0.87}$ | $58.45_{\pm1.01}$ | $\underline{57.67}_{\pm0.76}$ | $55.43_{\pm0.86}$ |
| IPR (Xiong et al., 2024) | $65.24_{\pm2.30}$ | $66.67_{\pm3.68}$ | $57.76_{\pm1.13}$ | $27.83_{\pm1.04}$ | $58.26_{\pm1.78}$ | $57.17_{\pm1.04}$ | $55.49_{\pm0.78}$ |
| HPL (Fixed-N(3)) | $69.52_{\pm1.48}$ | $\mathbf{74.38}_{\pm1.14}$ | $\underline{60.21}_{\pm2.04}$ | $\mathbf{30.17}_{\pm1.61}$ | $58.75_{\pm0.67}$ | $\underline{57.67}_{\pm0.58}$ | $\underline{58.45}_{\pm0.60}$ |
| HPL (Fixed-K(3)) | $70.48_{\pm1.09}$ | $66.42_{\pm2.69}$ | $58.34_{\pm1.84}$ | $28.33_{\pm0.58}$ | $\underline{59.69}_{\pm0.58}$ | $57.17_{\pm0.76}$ | $56.74_{\pm0.79}$ |
| HPL (Uncertainty) | $\mathbf{74.53}_{\pm2.89}$ | $64.18_{\pm1.29}$ | $58.75_{\pm0.55}$ | $27.83_{\pm0.76}$ | $59.11_{\pm0.66}$ | $57.33_{\pm0.29}$ | $56.95_{\pm0.44}$ |
| HPL (Semantic) | $\underline{72.86}_{\pm1.89}$ | $\underline{74.13}_{\pm1.88}$ | $\mathbf{60.74}_{\pm1.08}$ | $\underline{30.00}_{\pm1.00}$ | $\mathbf{60.39}_{\pm0.74}$ | $\mathbf{58.50}_{\pm1.00}$ | $\mathbf{59.44}_{\pm0.63}$ |
| **Qwen2.5-7B-Instruct** | 38.57 | 45.52 | 56.61 | 19.50 | 8.80 | 8.50 | 29.58 |
| SFT | $67.62_{\pm2.18}$ | $73.63_{\pm3.11}$ | $60.64_{\pm1.12}$ | $31.83_{\pm1.26}$ | $66.70_{\pm1.11}$ | $65.17_{\pm0.76}$ | $60.93_{\pm0.71}$ |
| RFT (Yuan et al., 2023) | $71.43_{\pm1.89}$ | $72.63_{\pm3.02}$ | $61.16_{\pm0.85}$ | $33.50_{\pm1.00}$ | $68.01_{\pm0.89}$ | $66.33_{\pm0.76}$ | $62.18_{\pm0.11}$ |
| ETO (Song et al., 2024) | $72.62_{\pm2.51}$ | $77.86_{\pm2.40}$ | $61.85_{\pm1.00}$ | $33.17_{\pm1.04}$ | $68.32_{\pm0.86}$ | $67.00_{\pm0.50}$ | $63.47_{\pm0.47}$ |
| IPR (Xiong et al., 2024) | $73.10_{\pm1.80}$ | $78.11_{\pm3.76}$ | $62.01_{\pm0.43}$ | $33.67_{\pm0.58}$ | $68.86_{\pm1.02}$ | $67.17_{\pm0.58}$ | $63.82_{\pm0.69}$ |
| HPL (Fixed-N(3)) | $78.33_{\pm2.51}$ | $78.86_{\pm2.40}$ | $62.11_{\pm0.41}$ | $34.33_{\pm0.76}$ | $\underline{69.55}_{\pm1.38}$ | $68.00_{\pm1.00}$ | $65.20_{\pm0.38}$ |
| HPL (Fixed-K(3)) | $\mathbf{85.71}_{\pm2.58}$ | $78.61_{\pm1.55}$ | $62.01_{\pm1.04}$ | $33.83_{\pm1.26}$ | $69.40_{\pm0.98}$ | $\underline{68.17}_{\pm0.58}$ | $66.29_{\pm0.54}$ |
| HPL (Uncertainty) | $\underline{83.10}_{\pm1.80}$ | $\underline{83.33}_{\pm1.88}$ | $\underline{62.79}_{\pm0.85}$ | $\mathbf{35.33}_{\pm1.04}$ | $69.21_{\pm0.47}$ | $67.83_{\pm0.29}$ | $66.93_{\pm0.43}$ |
| HPL (Semantic) | $82.62_{\pm2.30}$ | $\mathbf{84.08}_{\pm2.28}$ | $\mathbf{62.97}_{\pm0.50}$ | $\underline{35.17}_{\pm0.58}$ | $\mathbf{70.37}_{\pm1.27}$ | $\mathbf{68.50}_{\pm1.32}$ | $\mathbf{67.28}_{\pm0.47}$ |

**RQ1.** Does HPL outperform strong baselines that rely on conventional learning granularities, such as the trajectory-level (ETO) and step-level (IPR)?

**RQ2.** What is the impact of different action group segmentation strategies on the final performance of the agent?

**RQ3.** How crucial is the dual-layer curriculum mechanism to the success of HPL, and what is the contribution of each curriculum layer?

**RQ4.** How do trajectory-level, step-level, and group-level losses contribute to HPL?

## 4.1 EXPERIMENTAL SETUP

We evaluate our proposed method, HPL, on three diverse and challenging long-horizon agent benchmarks: ALFWorld (Shridhar et al., 2021), WebShop (Yao et al., 2022), and InterCode-SQL (Yang et al., 2023). In all our benchmarks, the environment returns a single terminal outcome reward at the end of each episode, which defines a finite-horizon episodic MDP and can be modeled with $\gamma$ close to 1 and suitably rescaled rewards. For all experiments, we use Qwen2.5-1.5B-Instruct and Qwen2.5-7B-Instruct as the backbone language models. HPL is compared against a suite of strong baselines, including SFT, RFT (Yuan et al., 2023), ETO (Song et al., 2024), and IPR (Xiong et al., 2024). All methods are initialized from an SFT model trained on expert trajectories generated by a GPT-4o teacher model. For MC estimation, we use $M = 5$ rollouts for per group. For Fixed-$N$ and Fixed-$K$ strategy, we set $N = 3$ and $K = 3$. For Uncertainty strategy, the entropy threshold $\epsilon$ is set to the 80th percentile of all action entropies (*i.e.*, the threshold for the top 20% highest values) computed across the training dataset. Detailed descriptions of the environments, baseline implementations, and all hyperparameters are deferred to Appendix D.

## 4.2 MAIN RESULTS (RQ1, RQ2)

As shown in Table 1, our HPL framework significantly outperforms all baseline methods across both model scales, providing a strong affirmative answer to our first research question. For the Qwen2.5-7B-Instruct model, our best-performing variant HPL (Semantic) achieves an average score of 67.28, surpassing the strongest single-granularity baselines ETO and IPR by substantial margins of 3.81 and 3.46 points, respectively. This trend of superiority holds true across all three diverse and challenging benchmarks. The advantage of HPL's hierarchical approach is particularly pronounced in tasks requiring complex, long-horizon generalization. For instance, on the ALFWorld benchmark's unseen scenarios, HPL (Semantic) achieves a mean success rate of 84.08%, a remarkable improve-

Table 2: Ablation study on our curriculum learning mechanism of HPL across three agent benchmarks.

| Models | ALFWorld | | WebShop | | InterCode-SQL | | Average |
|---|---|---|---|---|---|---|---|
| | seen | unseen | avg. reward | success rate | avg. reward | success rate | |
| **Qwen2.5-1.5B-Instruct** | | | | | | | |
| HPL | **71.43** | **72.39** | **59.99** | **30.00** | 60.08 | **58.50** | **58.73** |
| HPL Static | 68.57 | 71.64 | 58.80 | 29.00 | 59.45 | 58.00 | 57.58 |
| HPL Length CL Only | 69.29 | **72.39** | 58.06 | 28.50 | 59.39 | 57.50 | 57.52 |
| HPL Difficulty CL Only | **71.43** | 70.71 | 58.83 | **30.00** | **60.50** | 58.00 | 58.25 |
| **Qwen2.5-7B-Instruct** | | | | | | | |
| HPL | **83.57** | **86.57** | **62.56** | **34.50** | **70.63** | **69.00** | **67.81** |
| HPL Static | 75.71 | 82.84 | 62.05 | 33.00 | 69.71 | 68.50 | 65.30 |
| HPL Length CL Only | 82.14 | 85.07 | 62.26 | 34.00 | 69.49 | 67.50 | 66.74 |
| HPL Difficulty CL Only | 81.43 | 85.82 | 62.27 | **34.50** | 69.63 | 68.00 | 66.94 |

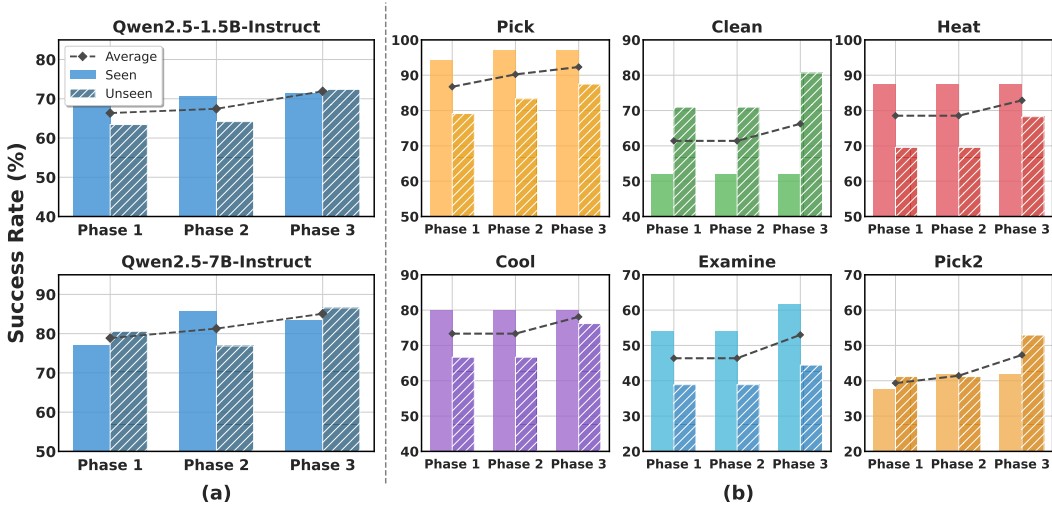

Figure 4: Phase-wise performance progression of HPL on the ALFWorld benchmark. **(a)** Success rates for both 1.5B and 7B models across the three curriculum phases. **(b)** A detailed breakdown for the 1.5B model on 6 sub-task types.

ment of nearly 6 points over the state-of-the-art IPR (78.11%). These results strongly indicate that by integrating preference signals at the trajectory, action, and group levels, HPL effectively resolves the granularity mismatch problem and learns a more robust and generalizable policy.

Our experiments also reveal the critical impact of the action group segmentation strategy, directly addressing our second research question. While all HPL variants consistently outperform the baselines, the adaptive, content-aware segmentation methods generally yield better performance than heuristic approaches. Notably, HPL (Semantic), which leverages a powerful language model to partition trajectories into semantically coherent sub-tasks, consistently emerges as the top-performing variant for both the 1.5B and 7B models. For the 7B model, it outperforms the next-best variant, HPL (Uncertainty), by over 0.35 point in the mean average score. This suggests that the quality of the action groups is paramount; providing the DPO loss with more meaningful, human-aligned sub-tasks as units of comparison leads to a more effective and powerful learning signal. Even so, the strong performance of the simpler Uncertainty and Fixed-N strategies demonstrates the inherent value of incorporating an intermediate granularity, validating the core design of our framework.

### 4.3 ANALYSIS OF THE CURRICULUM LEARNING MECHANISM (RQ3)

**Ablation of Curriculum Components.** To quantitatively assess the importance of our dual-layer curriculum, we conduct an ablation study by removing each layer individually, with results presented in Table 2. The primary finding is that the full HPL model, equipped with both curriculum layers, consistently outperforms all ablated variants. Removing the curriculum entirely (HPL Static) results in the most significant performance degradation across both model scales, confirming that employ-

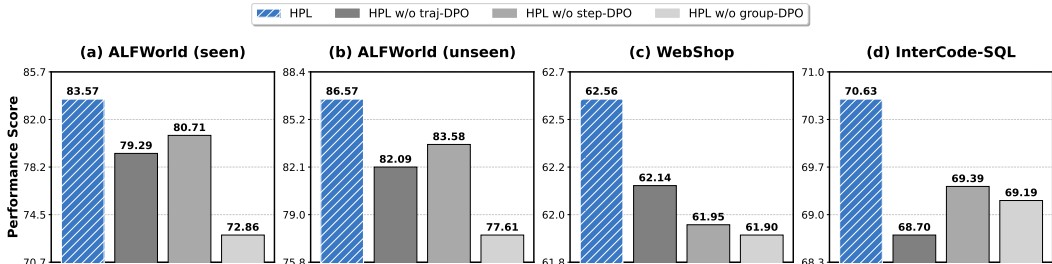

Figure 5: Ablation study on the HPL loss components on Qwen2.5-7B-Instruct.

ing a curriculum is crucial for effective learning. For the 7B model, HPL Static trails the full HPL by 2.51 average points, a substantial margin that underscores the necessity of a structured learning progression. Furthermore, the results indicate that both the length and difficulty-based curricula contribute positively to the final performance, with their individual removal leading to noticeable performance drops. This demonstrates the synergistic benefit of our dual-layer design, where organizing the learning process first by task complexity (length) and then by solution quality (difficulty) provides a more effective path to mastering complex agent behaviors.

**Phase-wise Performance Progression.** To provide a more fine-grained view of how the curriculum works, Figure 4 visualizes the agent's performance at the end of each curriculum phase. Panel (a) shows a clear improvement in the overall success rate for both the 1.5B and 7B models as they progress from Phase 1 to Phase 3. This trend holds for the unseen scenarios, indicating that the curriculum effectively helps the model generalize its learned skills. Panel (b) offers a deeper insight by breaking down the performance by sub-task type for the 1.5B model. We observe that while simpler tasks like Pick are learned relatively early, more complex tasks requiring longer reasoning chains, such as Clean and Pick2, show the most substantial performance gains in the later phases. This stage-wise improvement, with harder tasks being mastered in later stages, provides qualitative evidence that our dual-layer curriculum successfully guides the agent through an easy-to-hard learning process, progressively building its capabilities for complex, long-horizon planning.

## 4.4 Ablation on Hierarchical DPO Losses (RQ4)

To investigate the individual contribution of each component in our hierarchical framework, we conduct an ablation study on the three DPO losses, with results shown in Figure 5. The results reveal that while all three loss components, trajectory, action, and group, contribute positively to the final performance, the group-level DPO is demonstrably the most critical. Across all benchmarks, removing the group-level signal (HPL w/o group-DPO) induces a significantly more severe performance degradation than removing either the trajectory or step-level signals. This provides compelling evidence that the action-group granularity is the primary driver of HPL's effectiveness, serving as a crucial bridge between coarse trajectory feedback and action supervision and validating the synergistic benefit of our three-level approach.

## 5 Conclusion

In this work, we address the critical issue of granularity mismatch in preference-based alignment for LLM agents. We introduce Hierarchical Preference Learning (HPL), a novel framework that resolves this challenge by integrating preference signals across three levels of abstraction: trajectory, action, and a crucial, intermediate action-group level. By partitioning trajectories into semantically coherent sub-tasks and optimizing a hierarchical DPO loss, HPL learns to directly prefer successful multi-step action sequences over flawed alternatives. Extensive experiments demonstrate that HPL outperforms strong prior methods that operate primarily at the extreme granularities of entire trajectories or individual actions, across a suite of complex, long-horizon agent benchmarks. In conclusion, our work underscores the importance of learning from hierarchical signals that mirror the compositional nature of complex tasks, paving the way for more capable LLM agents.

## REPRODUCIBILITY STATEMENT

Our experimental setup is briefly summarized in Section 4.1. To facilitate the reproduction of our results, Appendix D offers a detailed account of all the benchmarks, data generation process, baselines, and all hyperparameters. We also release the code and data, which are available at: `https://github.com/PatrickG1014/HPL`.

## ACKNOWLEDGMENTS

This work is supported in part by National Natural Science Foundation of China (No. 62472427 and No. 62422215), Major Innovation & Planning Interdisciplinary Platform for the "Double First-Class" Initiative, Renmin University of China, Public Computing Cloud, Renmin University of China, fund for building world-class universities (disciplines) of Renmin University of China, and in part by the Open Research Fund from Guangdong Laboratory of Artificial Intelligence and Digital Economy (SZ) under Grant No. GML-KF-24-33.

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

## A    THE USE OF LLMs

In the preparation of this manuscript and the development of our codebase, we utilized LLMs to assist in two primary capacities. We detail these uses below for full transparency:

- **Writing and Editing Assistance:** We used LLM as a writing assistant to improve the grammar, phrasing, and overall clarity of the manuscript. The authors directed all core ideas and claims, and are fully responsible for the final wording, arguments, and content presented in this paper.
- **Code Implementation Support:** During the implementation of our experimental framework, an LLM served as a coding assistant for tasks such as generating basic code and aiding in debugging. The authors designed the overall software architecture, and all LLM-generated code was carefully reviewed and adapted by the authors to ensure its correctness and functionality.

## B    BROADER IMPACTS AND LIMITATIONS

**Broader Impacts.**    Our work on Hierarchical Preference Learning (HPL) presents a more efficient and effective paradigm for training capable autonomous agents from fixed datasets after a one-shot exploration phase. Technologically, this reduces the reliance on costly and often unsafe online exploration, making the development of sophisticated LLM agents more accessible and sustainable. The ability to learn from the compositional structure of tasks via action groups could lead to more reliable and predictable agents in real-world applications, from personalized assistants to complex workflow automation. On a societal level, the deployment of more competent agents can enhance productivity and assist with complex decision-making. However, as agent capabilities advance, ensuring their alignment with human values becomes paramount. The preferences learned from static datasets must be carefully curated to prevent the codification of biases or unintended behaviors. Furthermore, the increasing autonomy of such agents necessitates continued research into robust safety protocols, transparency, and ethical oversight to mitigate potential misuse and ensure these powerful technologies are developed responsibly for the benefit of society.

**Limitations.**    While our HPL framework demonstrates significant performance gains, we acknowledge several limitations that offer avenues for future research. Firstly, the effectiveness of HPL, particularly the semantic variant, is contingent on the quality of the action group segmentation. Although our experiments show that simple heuristics are beneficial, a suboptimal segmentation could yield less meaningful sub-tasks and hinder the learning process. Secondly, our dual-layer curriculum, while effective, introduces a new set of hyperparameters for defining task complexity and sample distinguishability, and the optimal configuration of this curriculum may be domain-specific and require careful tuning. Finally, our current work relies on a powerful teacher model for generating preference data, which means the agent's policy may inherit biases from the teacher. Future research could explore more robust, self-supervised segmentation techniques and investigate methods for learning curricula directly from the data.

## C    DISCUSSION

In this section, we discuss why group-level objectives are effective for long-horizon LLM agents. Building on both our theoretical analysis and empirical ablations, we highlight two complementary perspectives: (i) segmentation introduces a useful **structural prior** over sub-tasks, and (ii) group-level objectives have favorable **statistical properties** in the finite-data, limited-rollout regime.

**Segmentation as a structural prior.** Group-level supervision implicitly encodes prior knowledge about long-horizon task structure: instead of treating every individual action as an equally important learning unit, it focuses the objective on short sub-trajectories that are more likely to correspond to meaningful sub-tasks. Although HPL reuses exactly the same interaction data as trajectory- and step-level baselines, reorganizing this data into action groups changes where supervision is concentrated. Empirical results show that the semantic variant using a stronger segmentation prior achieves the best overall performance, while simpler segmenters still improve over both trajectory and step-level DPO, despite having almost no semantic information about sub-task boundaries.

**Statistical properties in the finite-data regime.** From an asymptotic perspective, a sufficiently expressive step-level DPO objective could in principle represent the same long-horizon credit assignment as a group-level objective. Our focus, however, is the practically relevant regime where both the dataset size and the MC rollout budget per state are limited. Under this regime, step-level MC estimates provide localized but noisy supervision: each decision point is updated from only a small number of rollouts. Group-level objectives alleviate this by aggregating multiple actions into a single supervised unit. With the same overall rollout budget, group-level rewards effectively average out noise over longer sub-trajectories and directly evaluate the joint contribution of several actions to task success. Proposition 1 formalizes this intuition through a bias–variance analysis, and our ablation studies further show that removing the group-level loss significantly degrades performance.

## D EXPERIMENTAL DETAILS

### D.1 ENVIRONMENTS AND TASKS

We evaluate our framework on three challenging benchmarks that require long-horizon, multi-step reasoning and interaction, representing a diverse set of agent tasks.

#### D.1.1 ALFWORLD

ALFWorld (Shridhar et al., 2021) is an embodied agent benchmark set in simulated household environments, uniquely designed to align abstract, text-based interactions with a visually rich, embodied world. Agents must parse natural language instructions and perform a sequence of high-level actions (*e.g.*, `goto`, `take`, `clean`) to complete common household tasks, such as "put a clean tomato on the sidetable". The benchmark is structured into six distinct and compositional sub-task categories: `Pick`, `Clean`, `Heat`, `Cool`, `Examine`, and `Pick2`. The dataset contains 3,553 scenarios for training. For evaluation, the test set is divided into a seen set of 140 scenarios to assess in-distribution generalization within familiar room layouts, and a more challenging unseen set of 134 scenarios with novel task instances in unobserved environments to evaluate out-of-distribution generalization. For all tasks, the maximum number of interaction turns is set to 30.

#### D.1.2 WEBSHOP

WebShop (Yao et al., 2022) is a web-based simulation environment that tasks agents with navigating a realistic e-commerce website to find and purchase a product that matches a given instruction. The environment is notable for its scale and realism, featuring 1.18 million real-world products and over 12,000 crowd-sourced, natural language instructions. To succeed, an agent must interact with multiple types of web pages, including search, results, and item-detail pages, by issuing high-level actions, primarily "`search[QUERY]`" or "`click[BUTTON]`". This process requires a combination of information retrieval skills to formulate effective search queries and long-horizon planning to navigate the site, compare items, and select the correct product options. The final reward is automatically calculated based on a heuristic that measures the attribute, option, and price match between the purchased item and the user's instruction. The benchmark includes 200 test tasks. For all tasks, the maximum number of interaction turns is set to 10.

#### D.1.3 INTERCODE-SQL

InterCode-SQL (Yang et al., 2023) is an interactive environment designed to benchmark agent capabilities in complex data querying tasks. Within a safe and reproducible Docker container, the agent interacts with a live MySQL database by iteratively writing and executing SQL queries to answer natural language questions. The environment is built upon the challenging, cross-domain Spider dataset (Yu et al., 2018), which requires the agent to understand complex database schemas and formulate queries that often involve multiple tables and `JOIN` operations. A key feature of this benchmark is its interactive nature; after each query execution, the agent receives real-world feedback, such as query results or error messages, which it must use to debug and refine its subsequent actions. A final reward is calculated based on the Intersection over Union (IoU) between the agent's submitted query results and the ground-truth records. The benchmark consists of 200 test tasks. For all tasks, the maximum number of interaction turns is set to 10.

## D.2 MODELS AND IMPLEMENTATION DETAILS

Our methodology relies on a dataset of expert trajectories and generating a corresponding set of suboptimal trajectories for creating preference pairs.

**Expert Trajectories ($\mathcal{D}_{\mathbf{expert}}$).**    Following prior work (Xiong et al., 2024), we generate our initial set of expert trajectories by prompting a powerful teacher model (GPT-4o) to solve tasks in each environment using a ReAct-style reasoning process. We then filter these generated trajectories, retaining only those that achieve a high outcome reward (*e.g.*, success score of 1.0 in ALFWorld and InterCode-SQL, or $>0.8$ in WebShop) to form our final expert dataset, $\mathcal{D}_{\text{expert}}$. This dataset is used for the initial behavior cloning stage.

**Models and Training.**    We utilize Qwen2.5-1.5B-Instruct and Qwen2.5-7B-Instruct as the backbone models for all experiments. For all DPO-based methods, including our HPL, the reference policy $\pi_{\text{ref}}$ is the SFT agent trained on $\mathcal{D}_{\text{expert}}$. During the behavior cloning phase, models are trained for 3 epochs using the AdamW optimizer with a cosine learning rate schedule, peaking at 1e-5. All experiments were conducted on 8 NVIDIA A800 80G GPUs.

## D.3 BASELINES

We compare HPL against a suite of strong baselines representing different alignment strategies, ranging from standard imitation learning to state-of-the-art preference optimization methods.

- **SFT**: The standard Supervised Fine-Tuning approach, where the model is trained only on the expert trajectories $\mathcal{D}_{\text{expert}}$. This serves as our foundational base model and represents the standard behavior cloning paradigm from which all other preference-based methods are initialized.
- **RFT** (Yuan et al., 2023): Rejection sampling Fine-Tuning is an enhanced fine-tuning method that uses rejection sampling to augment the expert dataset with newly generated successful trajectories. By enriching the training data with a more diverse set of successful paths, RFT serves as a strong imitation learning baseline that tests the performance limits of learning solely from positive demonstrations.
- **ETO** (Song et al., 2024): A trajectory-level DPO baseline that learns by contrasting full successful and failed trajectories. This method represents the coarsest end of the preference learning spectrum, providing a holistic signal based on the final outcome. While powerful, this approach faces challenges in credit assignment, as it can struggle to pinpoint specific errors within long action sequences.
- **IPR** (Xiong et al., 2024): A state-of-the-art process supervision method that performs step-level DPO using rewards estimated from Monte Carlo rollouts. IPR operates at the finest granularity, providing precise, localized feedback on individual actions. However, this step-level focus may overlook the synergistic value of multi-step sub-tasks.

## D.4 HYPERPARAMETERS

Table 3 and Table 4 show the hyperparameters for SFT and Group-DPO stage respectively across three agent benchmarks. All experiments were conducted on 8 NVIDIA A800 80G GPUs.

## D.5 RESOURCE COMPARISON

We report in Table 5 a resource comparison of SFT, ETO, IPR, and the HPL variants on ALFWorld with Qwen2.5-1.5B-Instruct, including whether an external powerful LLM is used, the number of LLM calls, and the generation/training time.

During data generation, we adopt the same parallel sampling implementation provided by the ETO and IPR codebases to accelerate environment interaction. The actual generation time depends primarily on the degree of parallelism in environment rollouts, as well as the time required to reset the

Table 3: Hyperparamenters for SFT stage across three agent benchmarks.

| Benchmark | ALFWorld | WebShop | InterCode-SQL |
|---|---|---|---|
| Batch size | 32 | 32 | 32 |
| Learning rate | 1e-5 | 1e-5 | 1e-5 |
| Optimizer | AdamW | AdamW | AdamW |
| LR scheduler | cosine | cosine | cosine |
| Warmup ratio | 0.1 | 0.1 | 0.1 |
| Max epochs | 3 | 3 | 3 |
| Max seq length | 6000 | 6000 | 6000 |
| DeepSpeed Zero stage | 3 | 3 | 3 |
| Gradient accumulation steps | 2 | 2 | 2 |

Table 4: Hyperparamenters for Group-DPO stage across three agent benchmarks.

| Benchmark | ALFWorld | WebShop | InterCode-SQL |
|---|---|---|---|
| Batch size | 32 | 32 | 32 |
| Learning rate | 3e-6 | 1e-6 | 1e-6 |
| $\beta$ | 0.3 | 0.3 | 0.3 |
| Optimizer | AdamW | AdamW | AdamW |
| LR scheduler | cosine | cosine | cosine |
| Warmup ratio | 0.1 | 0.1 | 0.1 |
| Max epochs | 1 | 1 | 1 |
| Max seq length | 6000 | 6000 | 6000 |
| DeepSpeed Zero stage | 3 | 3 | 3 |
| Group length ($L$) threshold for curriculum | (0,3,6) | (0,2,4) | (0,2,4) |
| Difficulty ($\Delta R$) threshold for curriculum | (1.0, 0.7, 0.4) | (1.0, 0.7, 0.4) | (1.0, 0.7, 0.4) |

environment and progress it to the desired states; the LLM call latency is not the dominant factor. The reported training times are based on real runs using 4 NVIDIA A800 80G GPUs.

## E ILLUSTRATIVE EXAMPLE OF GROUP REWARD ESTIMATION

In this section, we provide a concrete walkthrough of how the Monte Carlo (MC) rollout mechanism estimates the expected outcome reward for a specific action group, as defined in Equation 2.

Consider an example from the ALFWorld benchmark:

- **Task Instruction:** `put a clean apple in fridge`
- **Current Context** ($c_i$)**:** The agent is located at the `sinkbasin 1` and is holding a dirty `apple 1`.
- **Candidate Action Group** ($G_i$)**:** The policy generates a coherent sequence of actions intended to complete the "cleaning" sub-task:

    `G_i = [go to sinkbasin 1, clean apple 1 with sinkbasin 1]`

To estimate the reward $\hat{r}(G_i)$, we first execute $G_i$. The simulation state transitions to a point where the agent is holding a *clean* apple at the sink. From this state, we perform $M = 3$ stochastic rollouts using the reference policy $\pi_{ref}$ to see if the task can be successfully completed.

- **Rollout 1** ($\tau^{(1)}$)**:** The agent successfully navigates to the fridge, opens it, and places the apple inside.
    $\rightarrow$ **Outcome Reward** $R(\tau^{(1)}) = 1.0$ (Success).
- **Rollout 2** ($\tau^{(2)}$)**:** The agent navigates to the fridge but attempts to place the apple without opening the fridge first. It fails to recover within the step limit.
    $\rightarrow$ **Outcome Reward** $R(\tau^{(2)}) = 0.0$ (Failure).

Table 5: Resource comparison of SFT, ETO, IPR, and HPL variants on the ALFWorld benchmark with Qwen2.5-1.5B-Instruct.

| Method | External powerful LLM | # LLM calls | Gen time | Train time |
|---|---|---|---|---|
| SFT | ✗ | 0 | 0 | 18min |
| ETO | ✗ | $\sim$ 30,000 | 1h 7min | 13min |
| IPR | ✗ | $\sim$ 750,000 (step-level part) | 6h 13min (step-level part) | 26min |
| **HPL (Fixed-N(3))** | ✗ | $\sim$ 207,000 (group-level part) | 3h 35min (group-level part) | 25min |
| **HPL (Fixed-K(3))** | ✗ | $\sim$ 213,000 (group-level part) | 4h 15min (group-level part) | 26min |
| **HPL (Uncertainty)** | ✗ | $\sim$ 194,000 (group-level part) | 3h 47min (group-level part) | 28min |
| **HPL (Semantic)** | ✓ | $\sim$ 221,000 (group-level part) | 3h 21min (group-level part) | 26min |

- **Rollout 3 ($\tau^{(3)}$):** The agent navigates to the fridge, opens it, and successfully places the apple.
  $\rightarrow$ **Outcome Reward** $R(\tau^{(3)}) = 1.0$ (Success).

Finally, the estimated reward for group $G_i$ is calculated as the average of these outcomes:

$$\hat{r}(G_i) = \frac{1}{3}(1.0 + 0.0 + 1.0) \approx 0.67.$$

This value $\hat{r}(G_i) = 0.67$ serves as the quality label for this action group. If paired with a lower-quality group (*e.g.*, one that failed to clean the apple), the difference $\Delta R$ determines the sample difficulty for our curriculum scheduler.

## F   DETAILS OF DUAL-LAYER CURRICULUM SCHEDULER

In this section, we provide the algorithmic implementation of our curriculum scheduler and a visual illustration of the phase-wise training progression.

Algorithm 1 outlines the logic for partitioning the preference data into the $3 \times 3$ grid based on Group Length ($L$) and Sample Difficulty ($\Delta R$), and selecting the active data subsets for the current training phase $s$. We utilize the hyperparameters specified in Appendix D.4.

---
**Algorithm 1** HPL Dual-Layer Curriculum Scheduler
---
1: **Input:** Dataset $\mathcal{D}_{all}$, Phase $s \in \{1, 2, 3\}$, Thresholds $T_L, T_{\Delta R}$.
2: **Output:** Training subset $\mathcal{D}_{train}^{(s)}$.
3: **Step 1: Data Partitioning**
4: Partition $\mathcal{D}_{all}$ into $3 \times 3$ buckets $\mathcal{B}_{l,d}$ where $l, d \in \{1, 2, 3\}$.
5: For each sample $x \in \mathcal{D}_{all}$, assign to $\mathcal{B}_{l,d}$ based on:
   - Length Level $l$: Determined by group length vs. $T_L$ (Short $\rightarrow$ 1, Long $\rightarrow$ 3).
   - Difficulty Level $d$: Determined by reward gap vs. $T_{\Delta R}$ (Easy $\rightarrow$ 1, Hard $\rightarrow$ 3).
6: **Step 2: Phase-based Selection**
7: Define the set of active bucket indices $\mathcal{I}^{(s)}$ for current phase $s$:
8: **if** $s = 1$ **then**
9:    $\mathcal{I}^{(1)} \leftarrow \{(1,1)\}$ {Phase 1: Foundational (Short & Easy)}
10: **else if** $s = 2$ **then**
11:    $\mathcal{I}^{(2)} \leftarrow \{(1,1), (1,2), (2,1)\}$ {Phase 2: Expansion}
12: **else**
13:    $\mathcal{I}^{(3)} \leftarrow \{(l,d) \mid 1 \le l, d \le 3\}$ {Phase 3: Full Scale}
14: **end if**
15: **return** $\mathcal{D}_{train}^{(s)} \leftarrow \bigcup_{(l,d) \in \mathcal{I}^{(s)}} \mathcal{B}_{l,d}$
---

## G   ANALYSIS

We now analyze the bias and variance of group-level DPO loss. Consider an MDP with discount factor $\gamma \in [0, 1)$. A trajectory $\tau$ of horizon $T$ is denoted as $\tau = (s_1, a_1, r_1, \ldots, s_T, a_T, r_T)$. Let $\pi_{\text{ref}}$ be a reference policy strictly positive on every state-action pair. For any sequence $u$, we define its discounted return with respect to the (unknown) optimal value function $V^*$ by

$$R(u) := \sum_{i \in u} \gamma^{i - t_0} r_i + \gamma^{|u|} V^*(s_{t_0 + |u|}), \tag{10}$$

where $t_0$ is the starting time index of $u$ and $|u|$ is its length. The true preference probability that $u_w$ is preferred to $u_l$ is modelled by the Bradley-Terry law

$$P(u_w \succ u_l) = \sigma \left( \beta \left[ \log \frac{\pi^*(u_w)}{\pi_{\text{ref}}(u_w)} - \log \frac{\pi^*(u_l)}{\pi_{\text{ref}}(u_l)} \right] \right) := \sigma(\beta \Delta^*), \tag{11}$$

where $\sigma(z) = \frac{1}{1+e^{-z}}$ and $\beta > 0$ is a fixed inverse-temperature. The population DPO loss for a generic distribution $\mu$ over pairs $(u_w, u_l)$ is

$$\mathcal{L}^\mu := -\mathbb{E}_\mu \log \sigma(\beta \Delta^*). \tag{12}$$

Given a dataset $\mathcal{D}$ of $N$ i.i.d. trajectories, each method forms its own empirical distribution $\mu_\bullet$ and minimizes

$$\mathcal{L}_\bullet(\theta; \mathcal{D}_\bullet) := -\frac{1}{|\mathcal{D}_\bullet|} \sum_{(u_w, u_l) \in \mathcal{D}_\bullet} \log \sigma(\beta \Delta_\theta), \tag{13}$$

where $\Delta_\theta := \log \frac{\pi_\theta(u_w)}{\pi_{\text{ref}}(u_w)} - \log \frac{\pi_\theta(u_l)}{\pi_{\text{ref}}(u_l)}$ and $\bullet \in \{\text{traj}, \text{step}, \text{group}\}$. We adopt the standard risk decomposition

$$\begin{aligned} \text{Risk}(\mathcal{L}_\bullet) &:= \mathbb{E}[(\mathcal{L}_\bullet - \mathcal{L}^{\mu\bullet})^2] \\ &= \text{Bias}(\mathcal{L}_\bullet)^2 + \text{Var}(\mathcal{L}_\bullet), \end{aligned} \tag{14}$$

where the expectation $\mathbb{E}[\cdot]$ is taken over the sampling distribution of $\mathcal{D}_\bullet$ and

$$\begin{aligned} \text{Bias}(\mathcal{L}_\bullet) &:= \mathbb{E}[\mathcal{L}_\bullet] - \mathcal{L}^{\mu\bullet}, \\ \text{Var}(\mathcal{L}_\bullet) &:= \mathbb{E}\left[ (\mathcal{L}_\bullet - \mathbb{E}[\mathcal{L}_\bullet])^2 \right]. \end{aligned} \tag{15}$$

**Proposition 1** (Bias-variance trade-off of group-level DPO loss). *Let $T$ denote the trajectory length, $\gamma \in [0, 1)$ the discount factor, and $R_{max}$ the maximum reward. Let $\mathcal{L}_{traj}$, $\mathcal{L}_{step}$, and $\mathcal{L}_{group}(k)$ denote the empirical losses of trajectory-level, step-level, and group-level DPO with group length $k < T$, respectively. Then there exists a constant $C > 0$ depending only on $(\gamma, \pi_{ref})$ such that for every $\epsilon \in (0, 1)$ the choice $k(\epsilon) = \left\lceil \log_\gamma \left( \frac{(1-\gamma)\epsilon}{2\beta R_{max}} \right) \right\rceil$ satisfies*

$$\text{Bias}(\mathcal{L}_{group}(k)) \leq \min\{\text{Bias}(\mathcal{L}_{traj}), \text{Bias}(\mathcal{L}_{step})\} + \epsilon, \tag{16}$$

$$\text{Var}(\mathcal{L}_{group}(k)) \leq \frac{C \log(1/\epsilon)}{T} \min\{\text{Var}(\mathcal{L}_{traj}), \text{Var}(\mathcal{L}_{step})\}. \tag{17}$$

*Proof.* First, we analyze the bias of the three losses. We compare the population losses induced by the three sampling schemes. Recall that the logistic loss is 1-Lipschitz, that is, for any scalar difference $z$,

$$|\log \sigma(z) - \log \sigma(z')| \leq |z - z'|. \tag{18}$$

Hence the bias of an empirical loss $\mathcal{L}_\bullet$ is controlled by

$$|\mathbb{E}[\mathcal{L}_\bullet] - \mathcal{L}^{\mu\bullet}| \leq \beta \mathbb{E}_{\mu_\bullet}[|\Delta_\theta - \Delta^*|], \tag{19}$$

which is governed by the error in the return difference induced by the length of the comparison unit. We now analyze the three cases: trajectory, step, and group. Trajectory-level DPO compares entire trajectories with no truncation. Hence $\text{Bias}(\mathcal{L}_{\text{traj}}) = 0$. Step-level DPO compares suffixes from time $t$ to $T$. Following the implementation in Xiong et al. (2024), these suffixes are not truncated either. Hence $\text{Bias}(\mathcal{L}_{\text{step}}) = 0$. For group-level DPO, the unit has fixed length $k < T$. For clarity, we define $R^*(t) := \sum_{i=t}^{T} \gamma^{i-t} r_i$. According to Bellman equation,

$$R^*(t) = \sum_{i=t}^{t+k-1} \gamma^{i-t} r_i + \gamma^k R^*(t+k). \tag{20}$$

Consider a group pair $(G_w, G_l)$ starting from the same state $s_t$, the true return difference should be $\delta_{\text{traj}} = R_w^*(t) - R_l^*(t)$, while group-level DPO uses $\delta_{\text{group}} = R(G_w) - R(G_l)$. Substituting Equation 20 into $\delta_{\text{traj}}$, we get

$$\delta_{\text{traj}} = \left[ \sum_{i=t, r_i \in G_w}^{t+k-1} \gamma^{i-t} r_i + \gamma^k R_w^*(t+k) \right] - \left[ \sum_{i=t, r_i \in G_l}^{t+k-1} \gamma^{i-t} r_i + \gamma^k R_l^*(t+k) \right] \tag{21}$$

$$= (R(G_w) - R(G_l)) + \gamma^k (R_w^*(t+k) - R_l^*(t+k)) \tag{22}$$

$$= \delta_{\text{group}} + \gamma^k (R_w^*(t+k) - R_l^*(t+k)). \tag{23}$$

Hence the error in the return difference is

$$|\delta_{\text{traj}} - \delta_{\text{group}}| = \gamma^k |R_w^*(t+k) - R_l^*(t+k)| \leq \gamma^k \cdot \frac{2R_{\max}}{1-\gamma}, \tag{24}$$

where the last step follows from the fact that the absolute value of any finite-horizon discounted sum is bounded by

$$\sum_{i=t+k}^{T} \gamma^{i-(t+k)} |r_i| \leq R_{\max} \sum_{j=0}^{T-(t+k)} \gamma^j < \frac{R_{\max}}{1-\gamma}. \tag{25}$$

Therefore, for a single preference pair, the error of group-level DPO loss satisfies

$$|\log \sigma(\beta \Delta_{\text{group}} - \log \sigma(\beta \Delta_{\text{traj}}))| \leq \beta |\Delta_{\text{group}} - \Delta_{\text{true}}| \leq \beta \cdot \frac{2R_{\max}}{1-\gamma} \gamma^k. \tag{26}$$

Taking the expectation, we get

$$\text{Bias}(\mathcal{L}_{\text{group}}) \leq \text{Bias}(\mathcal{L}_{\text{traj}}) + \frac{2\beta R_{\max}}{1-\gamma} \gamma^k. \tag{27}$$

By choosing $k(\epsilon) = \left\lceil \log_\gamma \left( \frac{(1-\gamma)\epsilon}{2\beta R_{\max}} \right) \right\rceil$, we obtain

$$\text{Bias}(\mathcal{L}_{\text{group}}(k)) \leq \epsilon = \min\{\text{Bias}(\mathcal{L}_{\text{traj}}), \text{Bias}(\mathcal{L}_{\text{step}})\} + \epsilon. \tag{28}$$

Next, we analyze the variance of the three losses. We derive element-wise bounds on the variance of the empirical loss

$$\mathcal{L}_\bullet = \frac{1}{|\mathcal{D}_\bullet|} \sum_{(u_w, u_l) \in \mathcal{D}_\bullet} \ell(\Delta_\theta), \quad \ell(\Delta_\theta) := -\log \sigma(\beta \Delta_\theta), \tag{29}$$

for $\bullet \in \{\text{traj}, \text{step}, \text{group}\}$. All samples are generated from $N$ i.i.d. trajectories.

For trajectory-level DPO, each trajectory contributes exactly one preference pair. The total number of samples $|\mathcal{D}_{\text{traj}}| = N$. The $N$ pairs are i.i.d., hence the covariance terms in the variance vanish:

$$\text{Var}(\mathcal{L}_{\text{traj}}) = \frac{1}{N^2} \sum_{i=1}^{N} \text{Var}\left[\ell(\Delta_\theta^{(i)})\right] = \frac{1}{N} \Sigma_{\text{traj}}, \tag{30}$$

where $\Sigma_{\text{traj}} := \text{Var}[\ell(\Delta_\theta)]$ under the trajectory-level sampling distribution.

For step-level DPO, from one trajectory we extract $T$ consecutive suffixes $(u_t)_{t=1}^{T}$ with $u_t = (s_t, \ldots, s_T)$. The total number of samples is $|\mathcal{D}_{\text{step}}| = NT$. However, the $T$ samples inside one trajectory are highly overlapped. Sequence $u_t$ and $u_{t+1}$ share $T - t - 1$ identical transitions. Therefore the covariance part in covariance is non-zero and large.

Since $|\Delta_\theta| \leq \frac{2R_{\max}}{1-\gamma}$, we have $0 \leq \ell(\Delta_\theta) \leq L_{\max} := \log(1 + e^{\frac{2\beta R_{\max}}{1-\gamma}})$. For any $t < s \leq T$, let $o = s - t$ (number of shared steps). The Cauchy-Schwarz inequality gives $\text{Cov}(\ell_t, \ell_s) \leq \gamma^o L_{\max}^2$. Summing over ordered pairs in one trajectory, we get

$$\sum_{1 \leq t < s \leq T} \text{Cov}(\ell_t, \ell_s) \leq L_{\max}^2 \sum_{o=1}^{T-1} (T-o)\gamma^o < L_{\max}^2 \frac{\gamma}{(1-\gamma)^2}. \tag{31}$$

We now consider total variance across $N$ trajectories. Each trajectory contributes $T$ samples, and samples from different trajectories are i.i.d. Hence,

$$\text{Var}(\mathcal{L}_{\text{step}}) = \frac{1}{(NT)^2}\left[ N \cdot T \cdot \text{Var}(\ell_t) + N \cdot 2 \sum_{1 \leq t < s \leq T} \text{Cov}(\ell_t, \ell_s) \right] \tag{32}$$

$$\leq \frac{L_{\max}^2}{NT} + \frac{2L_{\max}^2 \gamma}{NT(1-\gamma)^2} = \frac{L_{\max}^2}{NT}\left(1 + \frac{2\gamma}{(1-\gamma)^2}\right). \tag{33}$$

$\text{Var}(\mathcal{L}_{\text{step}})$ is $O(\frac{1}{NT})$ but with a constant that does not degrade with $T$.

For group-level DPO, we extract $M = \lfloor T/k \rfloor$ non-overlapping groups of length $k$. The total number of samples is $|\mathcal{D}_{\text{group}}| = NM$. Between-trajectory samples are i.i.d., while within-trajectory samples are independent by construction. Therefore, the covariance terms in variance are zero, and

$$\text{Var}(\mathcal{L}_{\text{group}}(k)) = \frac{1}{(NM)^2} \cdot NM \cdot \text{Var}[\ell(\Delta_\theta)] = \frac{1}{NM}\Sigma_{\text{group}}, \tag{34}$$

where $\Sigma_{\text{group}} := \text{Var}[\ell(\Delta_\theta)]$ under the group-level distribution. Since a sub-trajectory has smaller variance than the full trajectory, $\Sigma_{\text{group}} \leq \Sigma_{\text{traj}}$. Inserting $M \geq T/k - 1$ into Equation 34, we obtain

$$\text{Var}(\mathcal{L}_{\text{group}}(k)) \leq \frac{k}{T} \cdot \frac{1}{N}\Sigma_{\text{traj}} = \frac{k}{T}\text{Var}(\mathcal{L}_{\text{traj}}). \tag{35}$$

An identical comparision with step-DPO gives

$$\text{Var}(\mathcal{L}_{\text{group}}(k)) \leq \frac{Ck}{T}\text{Var}(\mathcal{L}_{\text{step}}), \quad C = \frac{2\text{tr}(\Sigma_{\text{step}})}{\text{tr}(\Sigma_{\text{group}})} \geq 1. \tag{36}$$

The constant $C$ depends only on $\gamma$ and $R_{\max}$, and is independent of $T$, $k$, and $N$.

With $k(\epsilon) = \left\lceil \log_\gamma \left( \frac{(1-\gamma)\epsilon}{2\beta R_{\max}} \right) \right\rceil = \Theta(\log(1/\epsilon))$, Equation 35 and 36 yield

$$\text{Var}(\mathcal{L}_{\text{group}}(k)) \leq \frac{C\log(1/\epsilon)}{T} \min\{\text{Var}(\mathcal{L}_{\text{traj}}), \text{Var}(\mathcal{L}_{\text{step}})\}, \tag{37}$$

which is the variance bound claimed in the proposition. $\square$

## H  ADDITIONAL EXPERIMENTS

### H.1  SUB-TASK PERFORMANCE ON ALFWORLD

To provide a more fine-grained analysis of our method's capabilities, we present a detailed breakdown of success rates on the six distinct sub-task types in ALFWorld for both seen (Table 6) and unseen (Table 7) sets. The largest performance gains are often observed in the complex sub-tasks, such as `Examine` and `Pick2`, which require longer reasoning chains.

### H.2  CASE STUDY

### H.2.1  ALFWORLD

Table 6: Sub-task success rate (%) comparison on the ALFWorld seen set.

| Models | Sub-task | | | | | | Overall |
| --- | --- | --- | --- | --- | --- | --- | --- |
| | Pick | Clean | Heat | Cool | Examine | Pick2 | |
| **Qwen2.5-1.5B-Instruct** | 8.57 | 0.00 | 0.00 | 0.00 | 0.00 | 0.00 | 2.14 |
| SFT | 88.57 | 44.44 | 62.50 | 72.00 | 46.15 | 41.67 | 62.14 |
| RFT (Yuan et al., 2023) | 88.57 | 40.74 | 62.50 | 72.00 | 46.15 | 41.67 | 61.43 |
| ETO (Song et al., 2024) | 91.43 | 40.74 | 62.50 | 72.00 | 46.15 | 41.67 | 62.14 |
| IPR (Xiong et al., 2024) | 94.29 | 44.44 | 68.75 | 72.00 | 46.15 | 41.67 | 64.29 |
| HPL (Fixed-N(3)) | **97.14** | 48.15 | 75.00 | 72.00 | 46.15 | **50.00** | 67.86 |
| HPL (Fixed-K(3)) | 94.29 | 51.85 | 75.00 | **80.00** | 46.15 | **50.00** | 69.29 |
| HPL (Uncertainty) | 94.29 | **55.56** | 87.50 | 80.00 | **61.54** | 50.00 | **72.86** |
| HPL (Semantic) | **97.14** | 51.85 | 87.50 | 80.00 | **61.54** | 41.67 | 71.43 |
| **Qwen2.5-7B-Instruct** | 74.29 | 29.63 | 37.50 | 32.00 | 15.38 | 16.67 | 38.57 |
| SFT | 94.29 | 51.85 | 75.00 | 72.00 | 53.85 | 41.67 | 67.14 |
| RFT (Yuan et al., 2023) | 97.14 | 55.56 | 75.00 | 80.00 | 61.54 | 54.17 | 72.86 |
| ETO (Song et al., 2024) | 94.29 | 55.56 | 87.50 | 72.00 | 53.85 | 45.83 | 70.00 |
| IPR (Xiong et al., 2024) | 94.29 | 59.26 | 87.50 | 80.00 | 61.54 | 45.83 | 72.86 |
| HPL (Fixed-N(3)) | 97.14 | 66.67 | 87.50 | 88.00 | 76.92 | 50.00 | 78.57 |
| HPL (Fixed-K(3)) | **100.00** | **88.89** | **93.75** | **92.00** | 84.62 | **66.67** | **88.57** |
| HPL (Uncertainty) | 97.14 | 74.07 | 87.50 | 88.00 | **92.31** | 50.00 | 81.43 |
| HPL (Semantic) | **100.00** | 77.78 | 87.50 | **92.00** | 76.92 | 58.33 | 83.57 |

Table 7: Sub-task success rate (%) comparison on the ALFWorld unseen set.

| Models | Sub-task | | | | | | Overall |
| --- | --- | --- | --- | --- | --- | --- | --- |
| | Pick | Clean | Heat | Cool | Examine | Pick2 | |
| **Qwen2.5-1.5B-Instruct** | 0.00 | 0.00 | 0.00 | 0.00 | 0.00 | 0.00 | 0.00 |
| SFT | 70.83 | 64.52 | 60.87 | 76.19 | 27.78 | 35.29 | 58.21 |
| RFT (Yuan et al., 2023) | 79.17 | 61.29 | 65.22 | 76.19 | 33.33 | 35.29 | 60.45 |
| ETO (Song et al., 2024) | 83.33 | 64.52 | 65.22 | 80.95 | 38.89 | 41.18 | 64.18 |
| IPR (Xiong et al., 2024) | 83.33 | 64.52 | 65.22 | 80.95 | 38.89 | 47.06 | 64.93 |
| HPL (Fixed-N(3)) | **87.50** | **80.65** | 73.91 | **85.71** | **44.44** | **52.94** | **73.13** |
| HPL (Fixed-K(3)) | 83.33 | 61.29 | 69.57 | 80.95 | 33.33 | 41.18 | 63.43 |
| HPL (Uncertainty) | 83.33 | 61.29 | 69.57 | **85.71** | 33.33 | 35.29 | 63.43 |
| HPL (Semantic) | **87.50** | **80.65** | 78.26 | 76.19 | **44.44** | **52.94** | 72.39 |
| **Qwen2.5-7B-Instruct** | 62.50 | 54.84 | 52.17 | 52.38 | 16.67 | 17.65 | 45.52 |
| SFT | 91.67 | 80.65 | 78.26 | 85.71 | 50.00 | 58.82 | 76.12 |
| RFT (Yuan et al., 2023) | 79.17 | 74.19 | 78.26 | 76.19 | 44.44 | 52.94 | 69.40 |
| ETO (Song et al., 2024) | 95.83 | 74.19 | 78.26 | 80.95 | 55.56 | 70.59 | 76.87 |
| IPR (Xiong et al., 2024) | 95.83 | 77.42 | 73.91 | 85.71 | 61.11 | 64.71 | 77.61 |
| HPL (Fixed-N(3)) | 95.83 | 77.42 | 73.91 | 80.95 | 55.56 | 64.71 | 76.12 |
| HPL (Fixed-K(3)) | 95.83 | 80.65 | 69.57 | 80.95 | 55.56 | 70.59 | 76.87 |
| HPL (Uncertainty) | **100.00** | **87.10** | 82.61 | 80.95 | 66.67 | **76.47** | 83.58 |
| HPL (Semantic) | **100.00** | **87.10** | 82.61 | **90.48** | **77.78** | **76.47** | **86.57** |

**Case Study of ALFWorld**

## IPR

**Task Instruction:** You are in the middle of a room. Looking quickly around you, you see a cabinet 6, a cabinet 5, a cabinet 4, a cabinet 3, a cabinet 2, a cabinet 1, a coffeemachine 1, a countertop 3, a countertop 2, a countertop 1, a drawer 3, a drawer 2, a drawer 1, a fridge 1, a garbagecan 1, a microwave 1, a shelf 3, a shelf 2, a shelf 1, a sinkbasin 1, a stoveburner 4, a stoveburner 3, a stoveburner 2, a stoveburner 1, and a toaster 1.
Your task is to: put a cool mug in cabinet.
**Thought:** I need to find a mug to put in the cabinet. The task description doesn't specify where the mug is, so I should look around the room to locate one.
**Action:** go to toaster 1
**Observation:** On the toaster 1, you see nothing.
**Thought:** The toaster is not where mugs would typically be found, but it's good to check all areas systematically. There's nothing on the toaster, so I'll continue searching.
**Action:** go to stoveburner 1
**Observation:** On the stoveburner 1, you see a pot 1.

*...... (search for the mug)*

**Thought:** Cabinet 3 has a vase, but no mug. I'll check cabinet 4.
**Action:** go to cabinet 4
**Observation:** On the cabinet 4, you see a mug 2.

**Thought:** Cabinet 4 has a mug. This could be the mug I need to put in the cabinet. I'll take it and then place it in the cabinet.
**Action:** take mug 2 from cabinet 4
**Observation:** You pick up the mug 2 from the cabinet 4.
**Thought:** With the mug in hand, I need to return to cabinet 1 to place the mug inside.
**Action:** go to cabinet 1
**Observation:** On the cabinet 1, you see a peppershaker 1, a soapbottle 1, and a vase 2.
**Thought:** To complete the task, I need to put the mug into the cabinet.
**Action:** put mug 2 in/on cabinet 1
**Observation:** You put the mug 2 in/on the cabinet 1.

## HPL

**Task Instruction:** You are in the middle of a room. Looking quickly around you, you see a cabinet 6, a cabinet 5, a cabinet 4, a cabinet 3, a cabinet 2, a cabinet 1, a coffeemachine 1, a countertop 3, a countertop 2, a countertop 1, a drawer 3, a drawer 2, a drawer 1, a fridge 1, a garbagecan 1, a microwave 1, a shelf 3, a shelf 2, a shelf 1, a sinkbasin 1, a stoveburner 4, a stoveburner 3, a stoveburner 2, a stoveburner 1, and a toaster 1.
Your task is to: put a cool mug in cabinet.
**Thought:** I need to find a mug to put in the cabinet. The sink basin is a common place where mugs are found after being used or washed.
**Action:** go to sinkbasin 1
**Observation:** On the sinkbasin 1, you see a cup 1, a dishsponge 2, a plate 2, and a potato 1.
**Thought:** There is no mug at the sink basin, but there are other items here. I should check the shelves as they often store mugs.
**Action:** go to shelf 1
**Observation:** On the shelf 1, you see a creditcard 3.

*...... (search for the mug)*

**Thought:** Cabinet 3 has a vase but no mug. I'll proceed to the next cabinet.
**Action:** go to cabinet 4
**Observation:** On the cabinet 4, you see a mug 2.

**Thought:** I found a mug in cabinet 4. I need to take it so I can put it in the cooler as instructed.
**Action:** take mug 2 from cabinet 4
**Observation:** You pick up the mug 2 from the cabinet 4.
**Thought:** Now that I have the mug, I need to cool it. The fridge is the best place to do this.
**Action:** go to fridge 1
**Observation:** The fridge 1 is closed.
**Thought:** The fridge is closed. I need to open it to cool the mug.
**Action:** cool mug 2 with fridge 1
**Observation:** You cool the mug 2 using the fridge 1.

*# The agent mistakenly believes that the task has been completed.*

**Thought:** The task is now complete, and I should prepare to move on to the next task or task set. I need to ensure that I approach each task with clarity and precision, adhering to the given instructions and using the appropriate tools and resources.

**Action:** prepare for next task

**Observation:** Nothing happens.

*...... (until the steps are exhausted)*

**Reward:** 0.0

---

**Thought:** The mug is now cool. I should return to the cabinet to complete the task.

**Action:** go to cabinet 1

**Observation:** On the cabinet 1, you see a peppershaker 1, a soapbottle 1, and a vase 2.

**Thought:** I am back at cabinet 1. I need to put the cool mug in the cabinet as per the task instructions.

**Action:** put mug 2 in/on cabinet 1

**Reward:** 1.0

## H.2.2 WEBSHOP

### Case Study of WebShop

#### IPR

**Task Instruction:** I am looking for a gluten free, 100% vegan plant based protein shake that is soy-free, and price lower than 40.00 dollars.

**Thought:** I need to find a plant-based protein shake that meets specific criteria including being gluten-free, 100% vegan, soy-free, and under $40. I'll start by searching with these details.

**Action:** search[gluten free, 100% vegan plant based protein shake that is soy-free]

**Observation:** Page 1 (Total results: 50) [SEP] Next > [SEP] B08FW5Y13G [SEP] Orgain Bundle - Chocolate and Vanilla Bean Protein Powder - ... [SEP] $53.98 [SEP] B0885Y2GD4 [SEP] OWYN 100% Vegan Plant-Based Protein Shake, Cookies & Creamless, 1 Count, with 20g Plant Protein, ... [SEP] $100.0 *...... (other product details)*

**Thought:** The first page of results doesn't seem to have any options that fit all my criteria. I should check the next page for more options.

**Action:** click[next >]

**Observation:** Page 2 (Total results: 50) [SEP] < Prev [SEP] Next > [SEP] B07DM8SP27 [SEP] OWYN 100% Vegan Plant-Based Protein Shake, Strawberry Banana, 24 Pack, 20g Plant Protein ... [SEP] $71.18 [SEP] B08H8VJJGZ [SEP] OWYN - 100% Vegan Plant-Based Pro Elite High Protein Shakes | Vanilla, 12 Fl Oz ... [SEP] $4.48 [SEP] *...... (other product details)*

#### HPL

**Task Instruction:** I am looking for a gluten free, 100% vegan plant based protein shake that is soy-free, and price lower than 40.00 dollars.

**Thought:** I need to find a plant-based protein shake that meets specific criteria including being gluten-free, 100% vegan, soy-free, and under $40. I'll start by searching with these details.

**Action:** search[gluten free, 100% vegan plant based protein shake that is soy-free]

**Observation:** Page 1 (Total results: 50) [SEP] Next > [SEP] B08FW5Y13G [SEP] Orgain Bundle - Chocolate and Vanilla Bean Protein Powder - ... [SEP] $53.98 [SEP] B0885Y2GD4 [SEP] OWYN 100% Vegan Plant-Based Protein Shake, Cookies & Creamless, 1 Count, with 20g Plant Protein, ... [SEP] $100.0 *...... (other product details)*

**Thought:** The first page of results doesn't seem to have any options that fit all my criteria. I should check the next page for more options.

**Action:** click[next >]

**Observation:** Page 2 (Total results: 50) [SEP] < Prev [SEP] Next > [SEP] B07DM8SP27 [SEP] OWYN 100% Vegan Plant-Based Protein Shake, Strawberry Banana, 24 Pack, 20g Plant Protein ... [SEP] $71.18 [SEP] B08H8VJJGZ [SEP] OWYN - 100% Vegan Plant-Based Pro Elite High Protein Shakes | Vanilla, 12 Fl Oz ... [SEP] $4.48 [SEP] *...... (other product details)*

**Thought:** On the second page, I see a product called "OWYN 100% Vegan Plant-Based Protein Shake" which seems to match most of my requirements except for the price. It's also listed as soy-free and vegan, and it's well below $40. I'll take a closer look at this option.
**Action:** click[b07dm8sp27]
**Observation:** < Prev [SEP] OWYN 100% Vegan Plant-Based Protein Shake, Strawberry Banana, 24 Pack, 20g Plant Protein with Omega-3, Prebiotic supplements, Superfoods Greens Blend, Gluten-Free, Soy-Free, Non-GMO [SEP] Price: $71.18 [SEP] Rating: [SEP] Description [SEP] Features [SEP] Reviews [SEP] Buy Now
**Thought:** This product has a high price compared to my budget, and although it's described as vegan and soy-free, the price is above $40. Since I'm looking for something within my budget, I might reconsider or look for another option that fits all my criteria including the price limit.

**Action:** click[buy now]
**Reward:** 0.5

**Thought:** On the second page, I see a product called "OWYN 100% Vegan Plant-Based Protein Shake" which seems to match my requirements. It's also well within my budget at $4.48. I should take a closer look at this option.
**Action:** click[b08h8vjjgz]
**Observation:** < Prev [SEP] OWYN - 100% Vegan Plant-Based Pro Elite High Protein Shakes | Vanilla, 12 Fl Oz | Dairy-Free, Gluten-Free, Soy Free, Keto Friendly, Zero Sugar, Vegetarian [SEP] Price: $4.48 [SEP] Rating: [SEP] Description [SEP] Features [SEP] Reviews [SEP] Buy Now
**Thought:** This product has all the features I'm looking for.

**Action:** click[buy now]
**Reward:** 1.0

### H.2.3 INTERCODE-SQL

**Case Study of InterCode-SQL**

**IPR**

**Task Instruction:** What is the money rank of the tallest poker player?
**Thought:** I need to check what tables are available to use.
**Action:** SHOW TABLES
**Observation:** [('people',), ('poker_player',)]
**Thought:** There are two tables available, 'people' and 'poker_player'. Since the question is asking about the money rank of the tallest poker player, I should check if there is a table related to poker players.
**Action:** DESC poker_player
**Observation:** [('Poker_Player_ID', 'int', 'NO', 'PRI', None, 'auto_increment'), ('People_ID', 'int', 'YES', 'MUL', None, ''), ('Final_Table_Made', 'double', 'YES', '', None, ''), ('Best_Finish', 'double', 'YES', '', None, ''), ('Money_Rank', 'double', 'YES', '', None, ''), ('Earnings', 'double', 'YES', '', None, '')]

**HPL**

**Task Instruction:** What is the money rank of the tallest poker player?
**Thought:** I need to check what tables are available to use.
**Action:** SHOW TABLES
**Observation:** [('people',), ('poker_player',)]
**Thought:** The table 'poker_player' might have information about the height and money rank of the players. I should check the structure of this table.
**Action:** DESC poker_player
**Observation:** [('Poker_Player_ID', 'int', 'NO', 'PRI', None, 'auto_increment'), ('People_ID', 'int', 'YES', 'MUL', None, ''), ('Final_Table_Made', 'double', 'YES', '', None, ''), ('Best_Finish', 'double', 'YES', '', None, ''), ('Money_Rank', 'double', 'YES', '', None, ''), ('Earnings', 'double', 'YES', '', None, '')]

**Thought:** The 'poker_player' table contains information about poker players, including their money rank. To find the money rank of the tallest poker player, I need to join the 'people' table with the 'poker_player' table on the 'People_ID' column. Then, I can order the results based on height and limit the result to 1 to get the tallest poker player's money rank.
**Action:**
```
SELECT Money_Rank
FROM people
JOIN poker_player ON
people.People_ID =
poker_player.Poker_Player_ID
ORDER BY height
LIMIT 1
```
**Observation:** [(141.0,)]
**Thought:** The query executed successfully and returned a single number, which is the money rank of the tallest poker player. This should be the answer to the question.

**Action:** submit
**Reward:** 0.0

**Thought:** The 'poker_player' table contains columns for money rank and height. I can write a SQL query to find the money rank of the tallest poker player.
**Action:**
```
SELECT Money_Rank
FROM poker_player
ORDER BY Height DESC
LIMIT 1
```
**Observation:** Error executing query: Unknown column 'Height' in 'order clause'.
**Thought:** Oops, it seems like there is no column named 'Height' in the 'poker_player' table. I should check the structure of the 'people' table to see if it has the height information.
**Action:** DESC people
**Observation:** [('People_ID', 'int', 'NO', 'PRI', None, 'auto_increment'), ('Nationality', 'text', 'YES', '', None, ''), ('Name', 'text', 'YES', '', None, ''), ('Birth_Date', 'text', 'YES', '', None, ''), ('Height', 'double', 'YES', '', None, '')]
**Thought:** The 'people' table does have a column for height. I can join the 'poker_player' and 'people' tables to get the money rank of the tallest poker player.
**Action:**
```
SELECT p.Money_Rank
FROM poker_player p
JOIN people pe ON p.People_ID
= pe.People_ID
ORDER BY pe.Height DESC
LIMIT 1
```
**Observation:** [(68.0,)]
**Thought:** The query executed successfully and returned the money rank of the tallest poker player as 68. This should be the answer to the question.

**Action:** submit
**Reward:** 1.0

# I PROMPTS

## I.1 ALFWORLD

**Instruction Prompt for ALFWorld**

Interact with a household to solve a task. Imagine you are an intelligent agent in a household environment and your target is to perform actions to complete the task goal. At the beginning of your interactions, you will be given the detailed description of the current environment and your goal to accomplish.

For each of your turn, you will be given the observation of the last turn. You should first think about the current condition and plan for your future actions, and then output your action in this turn. Your output must strictly follow this format:"Thought: your thoughts.\nAction: your next action".

The available actions are:

1. go to {recep}

2. take {obj} from {recep}

3. put {obj} in/on {recep}

4. open {recep}

5. close {recep}

6. toggle {obj} {recep}

7. clean {obj} with {recep}

8. heat {obj} with {recep}

9. cool {obj} with {recep}

where {obj} and {recep} correspond to objects and receptacles.

After your each turn, the environment will give you immediate feedback based on which you plan your next few steps. if the envrionment output "Nothing happened", that means the previous action is invalid and you should try more options.

Your response should use the following format:

Thought: <your thoughts>
Action: <your next action>

---

## Semantic Grouping Prompt for ALFWorld

I need you to help me divide the trajectory of an agent's interaction with the environment into multiple action groups based on semantic relevance.

Below is an interaction trajectory, which contains the environment description received by the agent and the sequence of actions performed:
{trajectory}

Please divide the action sequence in this trajectory into multiple semantically related groups, each group represents a set of actions to complete a sub-goal or sub-task.
Please follow the following principles when dividing:

1. Actions in the same group should be semantically closely related and complete a clear subtask together

2. When the purpose of an action changes, it should be divided into a new group

3. For each group, briefly describe the common goal of the group of actions

Please use the following format to return the results:

<action_groups>
Group 1 (action index: 0-2): Find the target item
- Action 0: go to toiletpaperhanger 1
- Action 1: go to toilet 1
- Action 2: take toiletpaper 1 from toilet 1
Group 2 (action index: 3-4): Complete the main task
- Action 3: go to toiletpaperhanger 1
- Action 4: put toiletpaper 1 in/on toiletpaperhanger 1
</action_groups>

## I.2 WEBSHOP

---

**Instruction Prompt for WebShop**

You are web shopping.
I will give you instructions about what to do.
You have to follow the instructions.
Every round I will give you an observation and a list of available actions, you have to respond an action based on the state and instruction.
You can use search action if search is available.
You can click one of the buttons in clickables.

An action should be of the following structure:
search[keywords]
click[value]

If the action is not valid, perform nothing.
Keywords in search are up to you, but the value in click must be a value in the list of available actions.
Remember that your keywords in search should be carefully designed.

Your response should use the following format:
Thought: I think ...
Action: click[something]

---

**Semantic Grouping Prompt for WebShop**

I need you to divide a sequence of actions into groups based on semantic relevance.

A possible grouping example:

Group 1 (action index: 0-0): Initial search phase
- Action 0: search[size 5 patent-beige high heel]

Group 2 (action index: 1-1): Preliminary screening and click to view product details
- Action 1: click[b09gxnyjcd]

Group 3 (action index: 2-3): Specification confirmation and detailed screening stage
- Action 2: click[beige-almond toe-patent leather]
- Action 3: click[5]

Group 4 (action index: 4-4): Purchase decision stage
- Action 4: click[buy now]

Your output then should be in the following format:
[[0, 0], [1, 1], [2, 3], [4, 4]]

Below is the interaction trajectory:
{trajectory}

Please group the actions by their indices. Your response MUST be a valid JSON array of arrays of integers, where each inner array represents a group of action indices.

Follow these rules STRICTLY:
1. Each action must belong to exactly one group.
2. The indices must be contiguous and cover the entire range from 0 to {num_actions} - 1.

---

3. The final output MUST NOT contain any text, explanations, code blocks, or markdown formatting outside of the JSON array itself. It should be a raw JSON string.
4. The last number in the last group MUST be {num_actions} - 1.

Example for a trajectory with 5 actions (indices 0, 1, 2, 3, 4):
[[0, 1], [2, 3], [4, 4]]

Another valid example:
[[0, 0], [1, 2], [3, 4]]

Your output must be only the JSON, like this:
[[0, 1], [2, 3], [4, 4]]

## I.3 INTERCODE-SQL

### Instruction Prompt for InterCode-SQL

You are a helpful assistant assigned with the task of problem-solving. To achieve this, you will interact with a MySQL Database system using SQL queries to answer a question.
At each turn, you should first provide your step-by-step thinking for solving the task. Your thought process should start with "Thought: ", for example: Thought: I should write a SQL query that gets the average GNP and total population from nations whose government is US territory.

After that, you have two options:

1) Interact with a mysql programming environment and receive the corresponding output. Your code should start with "Action: " and should be surrounded with ```sql``` tag, for example:

Action:

```sql
SELECT AVG(GNP), SUM(population)
FROM nations
WHERE government = 'US Territory';
```

2) Directly submit the result, for example: Action: submit.

You should use this format: "Thought: your thought\nAction: \n```sql\n<the mysql command>\n```". You will receive the corresponding output for your sql command.
Your output should contain only one "Action" part.
The "Action" part should be executed with a mysql interpreter or propose an answer. Any natural language in it should be commented out.
The SQL query and submit parts can not appear in your output simutaneously.

### Semantic Grouping Prompt for InterCode-SQL

I need you to divide a sequence of actions into groups based on semantic relevance.

A possible grouping example:

Group 1 (action index: 0-1): Task initialization and data structure exploration phase
- Action 0: SHOW TABLES
- Action 1: DESC university

Group 2 (action index: 2-2): Query construction and execution phase
- Action 2: `SELECT Enrollment, Primary_conference FROM university ORDER BY Founded ASC LIMIT 1`

Group 3 (action index: 3-3): Result confirmation and submission stage
- Action 3: submit

Your output then should be in the following format:
[[0, 1], [2, 2], [3, 3]]

Below is the interaction trajectory:
{trajectory}

Please group the actions by their indices. Your response MUST be a valid JSON array of arrays of integers, where each inner array represents a group of action indices.

Follow these rules STRICTLY:
1. Each action must belong to exactly one group.
2. The indices must be contiguous and cover the entire range from 0 to {num_actions} - 1.
3. The final output MUST NOT contain any text, explanations, code blocks, or markdown formatting outside of the JSON array itself. It should be a raw JSON string.
4. The last number in the last group MUST be {num_actions} - 1.

Example for a trajectory with 5 actions (indices 0, 1, 2, 3, 4):
[[0, 1], [2, 3], [4, 4]]

Another valid example:
[[0, 0], [1, 2], [3, 4]]

Your output must be only the JSON, like this:
[[0, 1], [2, 3], [4, 4]]

