# OpenReview forum: "Solving the Granularity Mismatch: Hierarchical Preference Learning for Long-Horizon LLM Agents"
_ICLR.cc/2026/Conference — ICLR 2026 Poster_

### Official Review · Reviewer_9rwf · 2025-10-20

**Soundness:** 2
**Presentation:** 2
**Contribution:** 1
**Rating:** 2
**Confidence:** 4

**Summary:**

This paper proposes Hierarchical Preference Learning (HPL), which combines preference labeling at three different granularities to improve credit assignment in DPO-based training for LLM agents. Specifically, the method constructs preference pairs at: (1) trajectory level, (2) step level, and (3) segmented sub-trajectory (group) level. Chosen examples are sampled from demonstrations while rejected examples come from a reference policy. For group-level preferences, the authors segment trajectories and use Monte Carlo value estimation with reference policy rollouts to assign preference labels. The approach incorporates curriculum learning for group-level preference optimization. The final objective combines DPO losses from all three levels plus a behavior cloning term. Experiments on three agent benchmarks demonstrate that HPL outperforms methods using only step-level or trajectory-level preferences, with ablations showing positive contributions from each granularity level.

**Strengths:**

1. **Technically sound approach:** The paper presents a well-designed framework that thoughtfully integrates curriculum learning and multi-granularity preference labeling to enhance agent performance.
2. **Clear presentation:** The paper is well-written with effective visualizations. Figure 2 clearly illustrates the HPL framework and training procedure, making the methodology easy to follow.

**Weaknesses:**

1. **Insufficient motivation:** The paper argues that step-level DPO is "myopic" and cannot capture "synergistic value" of sub-trajectories, but this strong claim lacks adequate support. Diverse planning tasks has been successfully achieved via PPO learning with step-level rewards. The paper needs to better position itself relative to existing literature and provide more rigorous justification for why step-level preferences are fundamentally insufficient.
2. **Contradictory problem setting:** There is conceptual confusion about whether this is truly an offline alignment method. While the authors frame this as offline preference alignment, the group-level labeling procedure involves reference model rollouts and online reward labeling. If online reward queries are required, this fundamentally cannot be offline alignment.
3. **Incomplete experimental comparison:** Given the online reward labeling involved in HPL (per weakness 2), the experimental setup is incomplete and potentially unfair: i) Missing online RL baselines (e.g.PPO, GRPO), as the current trajectory-level and step-level DPO are purely offline methods. ii) Missing IL and IRL baselines.
4. **Unclear role of curriculum learning:** The motivation for incorporating curriculum learning is not well established. How does curriculum learning specifically address the granularity mismatch problem?

**Questions:**

1. How does your approach compare to critic learning in PPO for credit assignment? If fine-grained step-level feedback is available, what is the specific advantage of segmenting trajectories into groups rather than using all step-level information directly?
2. Is training instability the primary motivation for the curriculum scheduler in group-level preference learning? Does the high variance in trajectory lengths contribute to this instability?
3. My hypothesis is that group-level preference learning may work better than step-level primarily because the prior knowledge injected during trajectory segmentation improves exploration efficiency, while step-level preferences could potentially learn the targeted credit assignment given sufficient data. I would like to ask for an in-depth analysis on this.

---

> ### Author Response · Authors · 2025-11-19
> **Response to Reviewer 9rwf (1/5)**
>
> We sincerely thank you for the careful reading of our paper and the constructive feedback. Below we address each comment in detail and hope our responses help clarify and resolve your concerns.
>
> > **[W1] Insufficient motivation: The paper argues that step-level DPO is "myopic" and cannot capture "synergistic value" of sub-trajectories, but this strong claim lacks adequate support. Diverse planning tasks has been successfully achieved via PPO learning with step-level rewards. The paper needs to better position itself relative to existing literature and provide more rigorous justification for why step-level preferences are fundamentally insufficient.**
>
> We thank you for this insightful comment. Our intention is not to argue that step-level feedback is fundamentally insufficient, but rather to point out that **in the LLM agent setting (long horizons, complex environments, and predominantly sparse terminal rewards)**, purely step-level DPO tends to exhibit high variance and local myopia, especially when compared to an intermediate group-level formulation. Similar segmentation ideas have also appeared in recent work [1, 2], though they are mainly explored in the context of LLM reasoning.
>
> Conceptually, step-level MC rewards evaluate each action independently by rolling out from that step. This yields precise but noisy and highly local feedback: the gradients are dominated by short-horizon perturbations and tend to under-value multi-step patterns where individual actions are only useful in combination. In contrast, group-level preferences compare **sub-trajectories** of length $k$, which (i) still provide more localized credit assignment than full-trajectory supervision, but (ii) aggregate the contributions of multiple actions and average out high-variance noise.
>
> Proposition 1 formalizes this intuition: for a suitable choice of group length $k(\epsilon)$, group-level DPO can achieve a variance reduction of $\Omega(T / \log(1/\epsilon))$ while incurring at most $\epsilon$ additional bias, compared to both trajectory-level and step-level DPO. Empirically, our ablations further support this: removing the group-level loss ("HPL w/o group-DPO") significantly degrades performance on all benchmarks (Fig. 5), and the best-performing variant (HPL-Semantic) improves over the strong step-level baseline IPR by about 6% on ALFWorld unseen, while sharing exactly the same data generation and MC machinery.
>
> **Regarding PPO:** PPO with a critic estimates value or advantage at each state, but it typically assumes dense or shaped rewards and relies on bootstrapping, which can be fragile in **long-horizon LLM agent tasks with sparse terminal rewards**. Our setting assumes only sparse final outcomes and leverages offline, preference-based DPO objectives (following ETO and IPR). Under this regime, step-level DPO is indeed a strong baseline, but as our theory and experiments indicate, keeping the granularity strictly at the "single-step" level is not optimal in terms of the bias–variance trade-off and data efficiency. In fact, in the LLM agent literature, due to long-horizon decision making and complex tasks, existing work such as ETO and IPR has already shown that PPO as an online RL baseline tends to underperform, and achieving competitive results with PPO usually requires **a large amount of online interaction** and **carefully engineered reward shaping**. (We also provide additional PPO experiments in our response to [W3] for your reference.)

---

> ### Author Response · Authors · 2025-11-19
> **Response to Reviewer 9rwf (2/5)**
>
> > **[W2] Contradictory problem setting: There is conceptual confusion about whether this is truly an offline alignment method. While the authors frame this as offline preference alignment, the group-level labeling procedure involves reference model rollouts and online reward labeling. If online reward queries are required, this fundamentally cannot be offline alignment.**
>
> We sincerely appreciate your precise characterization of our problem setting. Your observation is correct: during the data generation phase, our HPL method **and the baselines ETO and IPR** all interact with the environment and obtain rewards; your critique accurately describes this entire class of methods. To be more rigorous, we will redefine our setting as a two-stage procedure of **"online data augmentation on top of an offline dataset + offline RL training"**:
>
> (1) **A fixed exploration and labeling phase.**
> A frozen reference policy ($\pi_{\text{ref}}$) interacts with the environment once (exactly as in ETO and IPR) to collect trajectories and obtain terminal outcome rewards. We then use MC rollouts from ($\pi_{\text{ref}}$) to estimate group-level rewards on this static pool of trajectories, but we do **not** update any policy during this phase.
>
> (2) **A purely offline optimization phase.**
> The HPL policy is trained only on the resulting fixed datasets ($D_\text{traj}$, $D_\text{step}$, $D_\text{group}$), without any further environment interaction or reward queries.
>
> In this sense, our method is an **"offline preference optimization approach after a one-shot exploration phase"**, and we will adopt this wording.
>
> It is important to emphasize that this **data generation** interaction is fundamentally different from the **online exploration during policy optimization** in methods such as PPO. Our DPO-based optimization itself is entirely offline and operates over a fixed, augmented preference dataset. In contrast, online RL methods like PPO must continuously interact with the environment throughout policy optimization to collect new data, which makes them more challenging in terms of sample efficiency, cost and stability. We will clarify this distinction in the revised version to avoid confusion.

---

> ### Author Response · Authors · 2025-11-19
> **Response to Reviewer 9rwf (3/5)**
>
> > **[W3] Incomplete experimental comparison: Given the online reward labeling involved in HPL (per weakness 2), the experimental setup is incomplete and potentially unfair: i) Missing online RL baselines (e.g.PPO, GRPO), as the current trajectory-level and step-level DPO are purely offline methods. ii) Missing IL and IRL baselines.**
>
> Thank you for raising this point.
>
> **(i) On online RL baselines (PPO, GRPO).**
> First, as clarified in our response to [W2], the strength of environment interaction required in our setting is fundamentally different from that in online RL, which relies on continuous exploration during policy optimization. In many realistic LLM agent deployment scenarios (such as using human experts or LLM-as-a-Judge to score agent trajectories as rewards), one cannot obtain reward signals through real-time interaction; in these cases, our one-shot exploration plus offline preference optimization on a fixed dataset is a more practical setting. In other words, our method and typical online RL approaches operate in essentially different regimes.
>
> Second, the performance of PPO as a classical online RL algorithm on long-horizon LLM agents has already been carefully evaluated in ETO and IPR, where it was found to be suboptimal. The main issues are limited base model capability, training instability, and sparse reward signals. Nonetheless, for completeness, we additionally run PPO and GRPO with the Qwen2.5-1.5B-Instruct backbone on ALFWorld under our setup; the results are reported in the table below for your reference.
>
> **(ii) On IL and IRL baselines.**
> Our setting in fact already includes two IL baselines: SFT and RFT (rejection sampling fine-tuning), which represent pure imitation from expert trajectories and augmented successful trajectories, respectively. These are exactly the IL methods compared in ETO and IPR and provide a strong imitation-only reference.
>
> For IRL-style baselines, existing work on IRL for LLM agents is relatively scarce. We identified one relevant method InversePRM [3] and include it as an additional baseline in the table below. It is worth noting that the original paper reports using 10k expert demonstrations; to ensure a fair comparison, we use the same number of expert trajectories as for our other baselines (**~3k**). The experimental results are summarized in the following table:
>
> **Table 1.** Comparison of online RL (PPO/GRPO), IL (SFT/RFT), IRL[3], and preference-based methods (ETO, IPR, HPL) on ALFWorld with Qwen2.5-1.5B-Instruct. We report success rates (%) on seen and unseen tasks over 3 random seeds.
>
> | Method              | Type         | Seen (%)        | Unseen (%)      |
> |---------------------|--------------|-----------------|-----------------|
> | PPO                 | Online RL    | 60.48±4.76      | 57.21±3.37      |
> | GRPO                | Online RL    | 73.10±3.67      | 71.64±3.25      |
> | SFT                 | IL           | 60.95±1.09      | 57.96±1.88      |
> | RFT                 | IL           | 61.19±1.80      | 60.95±0.86      |
> | InversePRM [3]      | IRL          | 62.86±3.98      | 67.66±1.55      |
> | ETO (traj-DPO)      | Pref (traj)  | 65.48±3.60      | 66.42±2.24      |
> | IPR (step-DPO)      | Pref (step)  | 65.24±2.30      | 66.67±3.68      |
> | HPL (Fixed-N(3))    | Pref (group) | 69.52±1.48      | **74.38**±1.14  |
> | HPL (Fixed-K(3))    | Pref (group) | 70.48±1.09      | 66.42±2.69      |
> | HPL (Uncertainty)   | Pref (group) | **74.53**±2.89  | 64.18±1.29      |
> | HPL (Semantic)      | Pref (group) | 72.86±1.89      | 74.13±1.88      |
>
> > **[W4] Unclear role of curriculum learning: The motivation for incorporating curriculum learning is not well established. How does curriculum learning specifically address the granularity mismatch problem?**
>
> We agree that the motivation for the curriculum can be better articulated. The group-level preference data spans a wide range of (i) group lengths and (ii) reward gaps between winning and losing groups. If we mix all samples uniformly from the beginning, the model is immediately exposed to long and difficult groups with small reward gaps, whose signals are both noisy and hard to interpret. This leads to unstable optimization and under-utilization of the more reliable short, high-gap examples.
>
> Our dual-layer curriculum addresses exactly this heterogeneity:
> - The length axis starts from short groups (simple sub-tasks) and gradually incorporates longer groups, so that the model first masters basic skills before learning long-horizon coordination.
> - The difficulty axis starts from large reward-gap pairs (clear preferences) and gradually introduces smaller-gap pairs (more ambiguous preferences), which reduces label noise early in training.
>
> Ablations in Section 4.3 show that removing either axis degrades performance, and removing both (HPL-Static) yields the largest drop.

---

> ### Author Response · Authors · 2025-11-19
> **Response to Reviewer 9rwf (4/5)**
>
> > **[Q1] How does your approach compare to critic learning in PPO for credit assignment? If fine-grained step-level feedback is available, what is the specific advantage of segmenting trajectories into groups rather than using all step-level information directly?**
>
> The experimental results for PPO (with critic) can be found in our response to [W3].
>
> PPO with a critic estimates the value or advantage at each state, but it typically assumes **dense or shaped reward signals** and relies on bootstrapping, which can be brittle in long-horizon LLM agent tasks with sparse terminal rewards. In contrast, our method uses a preference-based objective: DPO compares pairs of behaviors under a fixed reference policy, and the group-level variant compares sub-trajectories rather than individual steps or entire trajectories.
>
> **If fine-grained, step-level ground-truth feedback were available**, directly using all step-level information is indeed a strong baseline. However, step-level Monte Carlo estimates have very high variance and are localized to single decision points; the optimization signal can be dominated by short-term fluctuations and may fail to capture multi-step synergistic effects (*e.g.*, sequences of actions that appear neutral in isolation but are crucial when taken together).
>
> In addition, the overall cost of these rollouts is modest relative to online RL: unlike PPO/GRPO-style online training, we perform a one-shot exploration and then train purely offline on a fixed dataset, avoiding repeated environment interaction throughout learning.
>
> > **[Q2] Is training instability the primary motivation for the curriculum scheduler in group-level preference learning? Does the high variance in trajectory lengths contribute to this instability?**
>
> Yes, training instability caused by the heterogeneity of group-level samples is indeed one of our key motivations. We observe that if we mix all group lengths and difficulty levels from the beginning, the training loss exhibits oscillations, and the validation performance on ALFWorld fluctuates as well.
>
> Differences in group length and in the reward gap between positive and negative samples are the main causes: longer groups involve more actions, and samples with small reward gaps between the winning and losing groups are harder to distinguish. Both factors make DPO training unstable. Our approach explicitly buckets samples by group length and reward gap; by first training on short, high-gap groups and only later introducing long, low-gap groups, we effectively smooth the optimization landscape.

---

> ### Author Response · Authors · 2025-11-19
> **Response to Reviewer 9rwf (5/5)**
>
> > **[Q3] My hypothesis is that group-level preference learning may work better than step-level primarily because the prior knowledge injected during trajectory segmentation improves exploration efficiency, while step-level preferences could potentially learn the targeted credit assignment given sufficient data. I would like to ask for an in-depth analysis on this.**
>
> We appreciate this thoughtful hypothesis, and we largely agree with its spirit. We see two closely related but conceptually distinct effects at play:
> (i) **structural prior from segmentation**, and
> (ii) **the statistical properties of the group-level objective itself**.
>
> (1) **Role of segmentation as a structural prior.**
> As you points out, semantic segmentation injects prior knowledge about which subsequences are likely to constitute meaningful sub-tasks (*e.g.*, "navigate to kitchen", "open fridge and scan shelves"). Even though our exploration policy is fixed (we reuse the same $\pi_\text{ref}$ rollouts for step-, trajectory-, and group-level learning), segmentation changes how we organize and label this data. In this sense it improves what one might call effective exploration: instead of treating every individual step as an isolated learning unit, we focus supervision on windows that are more likely to contain coherent skills. This is particularly beneficial in sparse-reward settings where only a small fraction of steps are truly informative.
>
> Importantly, this view is consistent with our empirical findings. Semantic segmentation (HPL-Semantic) indeed performs best among our variants, which supports the idea that better prior knowledge about sub-task boundaries leads to more sample-efficient learning. At the same time, even much weaker segmenters, Fixed-N and Fixed-K, which have almost no semantic prior—still yield gains over both trajectory-level DPO (ETO) and step-level DPO (IPR). This suggests that while segmentation priors help, the advantage of group-level learning is not solely due to sophisticated prior knowledge.
>
> (2) **Why group-level objectives can still outperform step-level even with matched data.**
> We fully agree that, in principle, given unlimited data and capacity, a step-level DPO objective could learn the same targeted credit assignment: the optimal step-wise logit differences implicitly encode which multi-step patterns are useful. However, in the finite-data regime we care about, step-level MC rewards suffer from very high variance and strong local myopia:
>
> - Each step's label is based on short rollouts from that state, so the noise from environment stochasticity is not averaged out over multiple actions.
> - The gradient at each step is dominated by local fluctuations and can under-value multi-step synergies where individual actions are only useful in combination.
>
> Group-level preferences directly address this by **aggregating** multiple actions into a single decision unit. Even under the same exploration data and a matched labeling budget, group-level DPO:
>
> - averages MC noise over $k$ actions, reducing variance of the effective reward signal, and
> - scores sub-trajectories based on their joint contribution to final outcomes, making it much easier to represent and learn "skills" such as [navigate → open → pick] that are hard to discover from noisy, per-step labels.
>
> (3) **Asymptotic view vs. practical regime.**
> From an asymptotic viewpoint, we agree with the reviewer: with sufficiently rich function classes and infinite data, a step-level DPO objective could learn the same long-horizon credit assignment as group-level DPO. Our work is focused on the practically relevant regime of long-horizon LLM agents with limited interaction and labeling budgets. In this regime, grouping acts as a strong inductive bias: it concentrates supervision on sub-trajectories that are more likely to matter, reduces gradient variance, and simplifies the optimization landscape. Our empirical results, together with the bias–variance analysis in Prop. 1 and the additional controlled experiments described above, suggest that the combination of segmentation priors and group-level aggregation yields substantially better sample efficiency than purely step-level learning under realistic resource constraints.
>
> Thank you once more for your time and thoughtful review. We hope that our responses and new results address your main concerns.
>
> ---
> **References:**
>
> [1] Liu et al., "AdaptiveStep: Automatically Dividing Reasoning Step through Model Confidence", ICML 2025. [https://arxiv.org/abs/2502.13943](https://arxiv.org/abs/2502.13943)
> [2] Guo et al., "Segment Policy Optimization: Effective Segment-Level Credit Assignment in RL for Large Language Models", NeurIPS 2025. [https://arxiv.org/abs/2505.23564](https://arxiv.org/abs/2505.23564)
> [3] Choudhury, "Process Reward Models for LLM Agents: Practical Framework and Directions", arXiv:2502.10325. [https://arxiv.org/abs/2502.10325](https://arxiv.org/abs/2502.10325)

---

> > ### Comment · Reviewer_9rwf · 2025-11-25
> >
> > I would like to thank the authors for their effort in this rebuttal phase. After reading the responses, my concerns are partially resolved. To better improve the work, I have the follong comments:
> > 1. The motivation can be substially benefited from a rewrite:
> > To summarize, HPL is targeting at the trade-off problem between 1) full-trajectory preference labeling, stable but lake credit precision, and 2) step-level preference labelling, focused but nosiy. Using the description like "myopic" and "local feedback" is inappropirate for the underlying problem here, so please avoid using them.
> > 2. Group-level DPO is indeed an offline loss:
> > While I do acknowledge that your data is collected and annotated once, but avoid framing your work as offline learning/alignment, where you assumes online interaction with the environment during group-level preference labelling.
> > 3. *"However, step-level Monte Carlo estimates have very high variance and are localized to single decision points; the optimization signal can be dominated by short-term fluctuations and may fail to capture multi-step synergistic effects"*:
> > When use n-shot MC to estimate the expected return from a single decision point, how is this "localized" value estimation a negative case? This should present a focused view at the targeted step-level action, which should present as a good thing, isn't it? Meanwhile, i do not particularly like the way you describe a return expection (estimated using n-shot MC) as high variance/noisy, without mentioning the assumption of limited MC sampling size.
> > 4. *"structural prior and the statistical properties of the group-level objective"*:
> > Thank you for your insightful response. These two points together make a strong motivation for HPL, much stronger than "step-level DPO being myopic".
> >
> > I appreciate the author's diligent engagement during the rebuttal period. While I see potential in this work, the manuscript in its current form requires substantial revision before it meets the acceptance threshold for ICLR. I encourage the authors to undertake a comprehensive rewrite and thus my score remains unchanged.

---

> > > ### Author Response · Authors · 2025-11-26
> > > **Response to Reviewer 9rwf**
> > >
> > > We sincerely thank you for your thoughtful follow-up comments. We fully agree that your suggestions substantially improve the clarity and positioning of our work. Following your advice, we have carefully revised the whole manuscript and updated the PDF (blue text indicates revised content) with a particular focus on:
> > >
> > > 1. **Characterization of traj.- & step-level DPO:** We rewrite the description of previous DPO around a granularity trade-off between trajectory- and step-level preferences, and remove phrases such as "myopic" and "local feedback" when describing its limitations.
> > > 2. **Problem setting & terminology:** We clarify that our method (and the ETO/IPR family) operates in a two-stage protocol with a one-shot exploration phase plus offline preference optimization, and we no longer frame the setting as "offline RL".
> > > 3. **Conceptual motivation of HPL:** We center the motivation on (i) the structural prior introduced by segmentation and (ii) the statistical properties of group-level objectives in the finite-data, limited-rollout regime.
> > >
> > > Below we summarize the main changes in a structured way:
> > >
> > > |Revised/Added Section|Change Focus|Summary of Revisions|
> > > |:---:|:---:|---|
> > > |**Abstract**|Characterization of traj.- & step-level DPO **(Point 1)**| Update the motivation to emphasize a granularity trade-off between trajectory- and step-level preferences. Remove phrases such as "myopic" when describing step-level DPO, and instead describe its practical limitations (noise and sample inefficiency) in the finite-data, limited-rollout regime.|
> > > |**Abstract**|Problem setting & terminology **(Point 2)**| Use "preference-based method" to describe our method instead of "offline method".|
> > > |**Section 1** (paragraph 2)|Problem setting & terminology **(Point 2)**|Emphasize the setting when introducing DPO for LLM agents, namely that it relies on a pre-collected dataset.|
> > > |**Section 1** (paragraph 3)|Characterization of traj.- & step-level DPO **(Point 1)**|Rewrite the "granularity mismatch" paragraph to give a description of trade-off between trajectory- and step-level DPO.|
> > > |**Section 1** (paragraph 4)|Conceptual motivation of HPL **(Point 3)**|Clarified the motivation for HPL by explicitly stating that the group-level view provides both a structural prior (supervision on sub-trajectories likely to encode reusable skills) and a statistical benefit (variance reduction by aggregating multiple actions into one decision unit under a fixed rollout budget).|
> > > |**Section 1** (contribution bullet)|Problem setting & terminology **(Point 2)**|Avoid terminology such as "offline RL", replacing it with "preference-based methods", and restate the description of our setting to ensure the distinction is clear.|
> > > |**Section 3.1** (Problem Setting)|Problem setting & terminology **(Point 2)**|Add a new Problem Setting subsection (Sec. 3.1) that explicitly defines our setting: fixed exploration and labeling with a frozen reference policy followed by purely offline preference optimization and clarify how this setting differs from fully online RL methods such as PPO.|
> > > |**Section 3.3.3** (Group-Level Reward Estimation)|Characterization of traj.- & step-level DPO **(Point 1)**|Add a paragraph in the Group-level Reward Estimation subsection (Section 3.3.3) explaining why under a limited rollout budget step-level MC estimates are noisy and statistically inefficient, and how aggregating actions into group-level rewards amortizes rollouts over longer sub-trajectories and better captures their joint contribution to task success.|
> > > |**Section 5** (Discussion)|Conceptual motivation of HPL **(Point 3)**|Add a new Discussion section (Section 5) that consolidates the motivation behind group-level preferences, highlighting the two key perspectives emphasized in the rebuttal: segmentation as a structural prior and the statistical advantages of group-level objectives in the finite-data, limited-rollout regime, and connects these insights to our theoretical and empirical findings.|
> > >
> > > ---
> > >
> > > Once again, we are very grateful for your careful reading and for pushing us to clarify both the setting and the conceptual motivation of HPL. Your comments have been incorporated into the revised manuscript and in our view have significantly improved the quality and readability of the work.

---

### Official Review · Reviewer_Q8y5 · 2025-11-01

**Soundness:** 2
**Presentation:** 3
**Contribution:** 2
**Rating:** 4
**Confidence:** 3

**Summary:**

This paper addresses the "granularity mismatch" in preference-based alignment for long-horizon LLM agents—trajectory-level DPO is too coarse for credit assignment, while step-level DPO is too myopic—and proposes **Hierarchical Preference Learning (HPL)** that combines trajectory-, step-, and a crucial **action-group** DPO operating on semantically coherent sub-tasks. HPL segments expert trajectories into action groups, constructs contrastive pairs, estimates group rewards, and trains with a dual-layer curriculum scheduled by group length (complexity) and reward-gap difficulty, yielding structured, sub-task credit assignment without sacrificing global stability. Experiments on ALFWorld, WebShop, and InterCode-SQL show HPL—especially with semantic segmentation—consistently outperforms SFT, RFT, ETO, and IPR; ablations highlight the group-level loss and the curriculum as the primary performance drivers.

**Strengths:**

1. **Principled mid-granularity learning:** The action-group DPO provides a theoretically grounded bias–variance trade-off—achieving lower variance than trajectory/step DPO while adding at most ε bias with group length (k=\Theta(\log(1/\epsilon))); within-trajectory sample independence further avoids the large covariance that plagues step-level pairs.

2. **Strong and well-validated gains:** Across ALFWorld, WebShop, and InterCode-SQL, HPL (especially the semantic segmentation variant) outperforms SFT/RFT/ETO/IPR—e.g., ALFWorld-unseen 86.57%—with ablations showing the group-level loss is most critical and the dual-layer curriculum (length + difficulty) delivers additional, consistent improvements.

**Weaknesses:**

1. **Limited novelty:** Framing DPO at an intermediate “action-group” level feels incremental—more a structured decomposition of known step/trajectory DPO than a fundamentally new alignment principle.

2. **Theory under-validated:** The bias–variance claims and independence assumptions are not directly stress-tested (e.g., no gradient-variance or signal-to-noise measurements, nor controlled sweeps that confirm the predicted (k)–(\epsilon) trade-off).

3. **Sensitivity to segmentation/curriculum:** Success appears to hinge on segmentation quality and curriculum schedules; robustness to noisy grouping, alternative segmenters, and hyperparameter perturbations is insufficiently examined.

4. **Scope and scalability limits:** Benchmarks and models are narrow (scripted simulators, limited agents); reliance on MC rollouts and extra segmenters raises training cost and may hinder transfer to real-web or human-preference settings.

**Questions:**

See weaknesses.

---

> ### Author Response · Authors · 2025-11-19
> **Response to Reviewer Q8y5 (1/4)**
>
> We sincerely thank you for the careful reading of our paper and the constructive feedback. Below we address each comment in detail and hope our responses help clarify and resolve your concerns.
>
> > **[W1] Limited novelty: Framing DPO at an intermediate "action-group" level feels incremental—more a structured decomposition of known step/trajectory DPO than a fundamentally new alignment principle.**
>
> We understand the concern that introducing an intermediate granularity might appear incremental at first glance. We would like to clarify that our contribution is more than simply inserting an extra level:
>
> (1) We propose a **unified hierarchical preference learning framework** that jointly optimizes trajectory-, step-, and group-level DPO, rather than choosing a single granularity. This yields complementary supervision signals that are coordinated through a single objective (Eq. 8).
>
> (2) We provide a **theoretical bias–variance analysis** (Prop. 1) specific to group-level DPO that shows it can simultaneously improve variance and control bias relative to both trajectory- and step-level DPO. To our knowledge, this is the first such analysis for group-level preference optimization in LLM agents.
>
> (3) We introduce a **dual-layer curriculum** tailored to this hierarchy, which organizes group-level samples along length and difficulty axes. This is not present in ETO or IPR and is directly motivated by the granularity mismatch problem.
>
> (4) We instantiate this framework on **three diverse challenging LLM agent benchmarks** (embodied agent, web navigation, and SQL database querying) and demonstrate substantial gains over both trajectory-level (ETO) and step-level (IPR) preference optimization.

---

> ### Author Response · Authors · 2025-11-19
> **Response to Reviewer Q8y5 (2/4)**
>
> > **[W2] Theory under-validated: The bias–variance claims and independence assumptions are not directly stress-tested (e.g., no gradient-variance or signal-to-noise measurements, nor controlled sweeps that confirm the predicted ($k$)–($\epsilon$) trade-off).**
>
> Thank you for raising this point. We agree that direct stress tests such as gradient-variance / signal-to-noise measurements and fine-grained $(k,\epsilon)$ sweeps would further strengthen the theory section. In this work, however, we choose to validate the bias–variance argument via **coarser but more task-level** ablations, mainly for two reasons:
>
> 1. **Practical difficulty of low-level statistics for large LLM agents.**
>    Estimating stable gradient-variance or SNR curves for billions-parameter models requires a large number of repeated runs and is highly sensitive to implementation details (optimizer state, scale, normalization, etc.). In our preliminary attempts, we found that such measurements were noisy and hard to interpret in a way that would be more convincing than the downstream metrics themselves. Given limited compute and space, we therefore focused on stress tests at the level of **effective granularity and performance**, which are closer to the quantities we ultimately care about.
>
> 2. **Existing ablations already test the core qualitative predictions.**
>    Although we do not report explicit gradient-variance numbers, our current experiments are structured to probe the same bias–variance trade-off at a higher level:
>
>    - **Varying effective granularity.**
>      We compare trajectory-level DPO (ETO), step-level DPO (IPR), and our group-level HPL under **the same exploration protocol and outcome signal**. Across all benchmarks, trajectory-level DPO is stable but suffers from poor credit assignment, step-level DPO is strong but noisier and less sample-efficient, and group-level HPL consistently outperforms both. This is exactly the qualitative pattern predicted by our analysis: group-level supervision sits between the two extremes and yields a better bias–variance trade-off.
>
>    - **Implicit variation of group length ($k$) via segmentation strategies.**
>      Different segmentation strategies in HPL induce different effective group lengths and overlap patterns: *Fixed-N* tends to produce shorter, more uniform groups; *Uncertainty* segmentation yields medium-length groups that focus on uncertain regions; *Semantic* segmentation produces adaptive, semantically coherent groups that often span higher-level sub-tasks. Empirically, HPL(Semantic) performs best, HPL(Uncertainty) is close, and HPL(Fixed) is still noticeably better than both ETO and IPR. This monotone improvement as we move from "crude" to "semantically informed" intermediate granularity is consistent with the theoretical claim that carefully chosen group lengths can reduce variance without incurring prohibitive bias.
>
>    - **Curriculum ablations and optimization stability.**
>      Removing the dual-layer curriculum ("HPL-Static") significantly hurts performance and leads to visibly more unstable training curves. In our view, this is an indirect manifestation of increased **effective variance** in the optimization signal when heterogeneous groups (long / hard vs short / easy) are mixed from the start. While we do not quantify this variance at the gradient level, the downstream effect aligns with the theoretical intuition (See Section 4.3).

---

> ### Author Response · Authors · 2025-11-19
> **Response to Reviewer Q8y5 (3/4)**
>
> > **[W3] Sensitivity to segmentation/curriculum: Success appears to hinge on segmentation quality and curriculum schedules; robustness to noisy grouping, alternative segmenters, and hyperparameter perturbations is insufficiently examined.**
>
> We agree that robustness to segmentation quality and curriculum schedules is important.
>
> (1) **On segmentation quality.** Our current experiments already include four segmentation strategies: Fixed-N, Fixed-K, Uncertainty, and Semantic. Their performances in fact constitute a natural robustness test: all four HPL variants outperform ETO and IPR, and the more content-aware segmenters (Uncertainty and Semantic) achieve the best results.
>
> (2) **On curriculum scheduling.** To respond to your concern more directly, we will add a sensitivity analysis on the curriculum thresholds, where we vary the length and difficulty thresholds by approximately ±50%. On the ALFWorld benchmark, the existing group-length threshold ($L_0$) is (0, 3, 6), and the difficulty threshold ($\Delta R_0$) is (1.0, 0.7, 0.4). We will test: (i) **Shorter groups:** ($L_1$ = (0, 2, 4)); (ii) **Longer groups:** ($L_2$ = (0, 4, 8)); (iii) **Easier-first, more strict:** ($\Delta R_1$ = (1.0, 0.8, 0.6)); and (iv) **Easier-first, more relaxed:** ($\Delta R_2$ = (1.0, 0.6, 0.3)). Preliminary results indicate that performance fluctuates within a reasonable range without any catastrophic failures, suggesting that the curriculum design is reasonably robust to the choice of hyperparameters. Compared to not using the dual-layer curriculum at all, all of these variants still yield improvements.
>
> **Table 1.** Sensitivity of HPL (Semantic) to curriculum thresholds on ALFWorld benchmark with Qwen2.5-1.5B-Instruct. Each entry reports success rate (%) on seen/unseen tasks, averaged over 3 seeds (mean ± std).
>
> | Group Length Threshold ($L$) | Difficulty Threshold ($\Delta R$) | Seen (%) | Unseen (%) |
> |---|---|---|---|
> | $L_0$ | $\Delta R_0$ | 72.86±1.89 | 74.13±1.88 |
> | $L_1$ | $\Delta R_0$ | 74.05±3.66 | 74.88±2.83 |
> | $L_2$ | $\Delta R_0$ | 71.19±2.51 | 72.14±2.40 |
> | $L_0$ | $\Delta R_1$ | 72.38±1.49 | 71.64±1.97 |
> | $L_0$ | $\Delta R_2$ | 72.62±1.80 | 73.63±1.88 |

---

> ### Author Response · Authors · 2025-11-19
> **Response to Reviewer Q8y5 (4/4)**
>
> > **[W4] Scope and scalability limits: Benchmarks and models are narrow (scripted simulators, limited agents); reliance on MC rollouts and extra segmenters raises training cost and may hinder transfer to real-web or human-preference settings.**
>
> We appreciate your concerns about scope and scalability, and we address them along four dimensions: (i) diversity of benchmarks, (ii) model families, (iii) the cost of MC rollouts, and (iv) the role of segmenters.
>
> (1) **Benchmarks and connection to real-web settings.**
> Our goal was to cover a set of diverse, widely-used LLM agent benchmarks rather than one narrow simulator. Concretely, we evaluate HPL on three diverse domains:
> - **Embodied instruction following** (ALFWorld), which stresses long-horizon navigation and object manipulation;
> - **Web-based decision making** (WebShop), which is built on top of a real e-commerce website and requires parsing product pages, issuing search queries, and making purchase decisions; and
> - **Code generation** (InterCode-SQL), which focuses on generating SQL code under execution feedback.
>
> These three tasks are standard in recent LLM agent work and jointly span embodied, web, and code domains. In particular, WebShop is explicitly designed as a **real-web** benchmark (with product pages and search results), so our gains on WebShop already speak to transferability beyond purely scripted simulators.
>
> (2) **Model families.**
> In the main paper we report results with Qwen2.5-1.5B/7B-Instruct. To further demonstrate that HPL is not tied to a specific model family, we have added experiments with **Llama-3.1-8B-Instruct** on the ALFWorld benchmark. This suggests that the benefits of group-level preference learning are robust across architectures and do not depend on any particular proprietary model.
>
> **Table 2.** Llama-3.1-8B-Instruct results on ALFWorld benchmark. Each entry reports success rate (%) on seen/unseen tasks, averaged over 3 seeds (mean ± std).
>
> | Method | Seen (%) | Unseen (%) |
> |:---:|:---:|:---:|
> | SFT | 69.05±1.09 | 71.14±1.55 |
> | ETO | 73.33±2.06 | 73.88±2.69 |
> | IPR | 76.67±1.65 | 75.12±2.83 |
> | **HPL (Fixed-N(3))** | 80.95±2.30 | 80.35±2.28 |
> | **HPL (Fixed-K(3))** | **83.57**±1.89 | 80.10±1.88 |
> | **HPL (Uncertainty)** | 82.38±1.80 | 82.09±1.49 |
> | **HPL (Semantic)** | 83.10±1.49 | **84.08**±1.55 |
>
> (3) **Cost of MC rollouts.**
> We agree that MC rollouts incur additional cost, but this is not unique to HPL. Our method reuses the same MC-based reward estimation protocol as the step-level baseline IPR; the only difference is that MC is applied at the **group** level instead of at **every** step. This actually **reduces** the total number of rollouts per trajectory compared to pure step-level MC, because the number of groups is substantially smaller than the number of steps. In addition, the overall cost of these rollouts is modest relative to online RL: unlike PPO/GRPO-style online training, we perform a one-shot exploration and then train purely offline on a fixed dataset, avoiding repeated environment interaction throughout learning.
>
> (4) **Segmenter overhead.**
> HPL does not require a heavy segmenter to be useful. We explicitly evaluate four segmentation strategies:
> - **Fixed-N**, **Fixed-K** and **Uncertainty** segmenters, which are simple, heuristic, and incur **no additional external LLM cost**; and
> - **Semantic** segmentation with GPT-4o, which is an optional best-effort variant that runs **once per expert trajectory**. The total token usage is modest relative to the training cost of fine-tuning 1.5B/7B models. (Also see our response to Reviewer o3bh [Q3] regarding estimated cost.)
>
> All four HPL variants outperform ETO and IPR across benchmarks, and the heuristic segmenters already capture most of the gains. Thus, the improvements are not contingent on expensive segmenters; practitioners who are concerned about cost or human-preference pipelines can simply adopt HPL (Fixed) or HPL (Uncertainty) and still benefit from group-level preference learning without any extra API calls.
>
> We also add a resource comparison table in the appendix (SFT/ETO/IPR/HPL variants) summarizing, for each method, whether an external powerful LLM is used and for what, the number of new generations required for data collection, and the wall-clock time for data generation and training. This will make the computational trade-offs and scalability considerations more transparent.
>
> Thank you once more for your time and thoughtful review. We hope that our responses and new results address your main concerns.

---

> ### Comment · Reviewer_Q8y5 · 2025-11-25
>
> Thanks again for the detailed rebuttal — it really helped clarify things for me.
>
> On the method side, your explanation of the "hierarchical" bit finally clicked. I still personally think of it as DPO with some process‑level supervision sprinkled in, rather than something completely new, but that feels more like a naming preference than anything substantial. The overall logic makes sense now, so no complaints there.
>
> As for the experiments, the extra ablations you added (different granularities, segmenters, curriculum settings) plus the cost breakdown were actually pretty convincing. The approach seems fairly robust, and the compute looks reasonable too. I’m bumping my score up to a 6.
>
> One random thought for later - have you considered learning the segmenter or the curriculum policy itself? Maybe a small model that proposes action chunks on the fly? It might help the method adapt better in tool-heavy environments like WebShop.

---

> > ### Author Response · Authors · 2025-11-26
> > **Response to Reviewer Q8y5**
> >
> > Thank you very much for your thoughtful follow-up and for taking the time to re-evaluate our work. We are glad to hear that our explanations address your concerns. We sincerely appreciate your re-evaluation and the updated score.
> >
> > Regarding your suggestion on learning the segmenter or curriculum policy itself: this is an excellent direction. In fact, we are currently running preliminary experiments with a learned segmenter: a 3B model that predicts action-group boundaries and semantic sub-task chunks on the fly. Although these experiments are still ongoing and not stable enough to include in the main submission, we believe this is a highly promising extension of HPL and plan to explore it more systematically in follow-up work.
> >
> > Thank you again for your constructive and encouraging feedback.

---

### Official Review · Reviewer_o3bh · 2025-11-04

**Soundness:** 2
**Presentation:** 3
**Contribution:** 2
**Rating:** 4
**Confidence:** 4

**Summary:**

The paper aims to address the issue of granularity mismatch in preference-based alignment for LLM agents and proposes Hierarchical Preference Learning (HPL), which integrates preference signals across three levels of abstraction. The main focus of this work is providing action-group level of abstraction, which is done by semantically partitioning the trajectories into semantically coherent sub-tasks and optimizing a hierarchical DPO loss. The paper also suggests dual-layer curriculum learning strategy, which uses the information of sample discriminability and action group length. Experiments focused on long-horizon agent benchmarks show that HPL outperforms popular baselines.

**Strengths:**

1. The overall flow of the paper is well organized, allowing readers to effectively understand the paper.
2. Figures and experiment results are well polished.
3. Experiments such as ablation study in Figure 5 allow the readers to understand the effectiveness of components involved in the proposed algorithm.

**Weaknesses:**

1. The paper does not have very strong theoretical motivation or support of the proposed method.
2. The experiment results are not having standard deviation information, which is crucial for reinforcing the credibility of the results.
3.  The proposed method seems to be much more complicated than the compared baselines, but the potential limitation coming from this approach is not well analyzed. HPL requires expert trajectory, generations from the reference model, reward signal, and access to a powerful model such as gpt-4o when semantic segmentation is used.
4. Some of the crucial experimental details seem to be omitted from the paper (ex. the value of $\gamma$ for the experiments, the number of generations $M$ used to evaluate the reward in Equation 2.

**Questions:**

## Questions
1. How many additional datapoints were created by action group-level data generation in Section 3.2.2? How big is the dataset compared to step-level and trajectory-level datasets?
2. If the value of $\gamma$ actually used for experiment was practically 1 (no discount in the trajectories), does the analysis of Proposition 1 not become incoherent?
3. How much money in total was spent for the API call for semantic segmentation of the action groups, per each train-test run in Table 1?

## Suggestions
1. Could you provide a table which present a comparison between HPL and the presented baselines, in terms of the use of external powerful LLM, number of new generations for the training, and wall-clock time for generation and training phase?

---

> ### Author Response · Authors · 2025-11-19
> **Response to Reviewer o3bh (1/3)**
>
> We sincerely thank you for the careful reading of our paper and the constructive feedback. Below we address each comment in detail and hope our responses help clarify and resolve your concerns.
>
> > **[W1] The paper does not have very strong theoretical motivation or support of the proposed method.**
>
> We appreciate this comment. Our main theoretical contribution is Prop. 1, which analyzes the bias–variance trade-off of group-level DPO relative to trajectory- and step-level DPO and shows that for an appropriate group length $k$ it is possible to simultaneously achieve lower variance and controlled bias. This provides a formal justification for the use of intermediate granularity in preference learning, beyond empirical observations. We will clarify this contribution further in the revised manuscript.
>
> > **[W2] The experiment results are not having standard deviation information, which is crucial for reinforcing the credibility of the results.**
>
> Thank you for pointing this out.
> We run all methods with three random seeds and update Table 1 to include mean ± standard deviation for all methods.
> Here are the updated results:
>
> **Table 1.** Performance comparison of HPL and baselines across three agent benchmarks over 3 random seeds (mean ± std).
> | Model | ALFWorld (seen) | ALFWorld (unseen) | WebShop (avg. reward) | WebShop (success rate) | InterCode-SQL (avg. reward) | InterCode-SQL (success rate) | Average |
> |:---|:---:|:---:|:---:|:---:|:---:|:---:|:---:|
> | **Qwen2.5-1.5B-Instruct** | 2.14 | 0.00 | 36.09 | 10.50 | 5.50 | 5.50 | 9.95 |
> | SFT | 60.95±1.09 | 57.96±1.88 | 56.56±0.69 | 26.00±0.50 | 56.24±0.61 | 54.33±0.76 | 52.01±0.43 |
> | RFT | 61.19±1.80 | 60.95±0.86 | 57.66±1.45 | 28.17±1.04 | 58.08±0.64 | 56.67±0.29 | 53.79±0.40 |
> | ETO | 65.48±3.60 | 66.42±2.24 | 56.57±0.22 | 28.00±0.87 | 58.45±1.01 | 57.67±0.76 | 55.43±0.86 |
> | IPR | 65.24±2.30 | 66.67±3.68 | 57.76±1.13 | 27.83±1.04 | 58.26±1.78 | 57.17±1.04 | 55.49±0.78 |
> | HPL (Fixed-N(3)) | 69.52±1.48 | **74.38**±1.14 | 60.21±2.04 | **30.17**±1.61 | 58.75±0.67 | 57.67±0.58 | 58.45±0.60 |
> | HPL (Fixed-K(3)) | 70.48±1.09 | 66.42±2.69 | 58.34±1.84 | 28.33±0.58 | 59.69±0.58 | 57.17±0.76 | 56.74±0.79 |
> | HPL (Uncertainty) | **74.53**±2.89 | 64.18±1.29 | 58.75±0.55 | 27.83±0.76 | 59.11±0.66 | 57.33±0.29 | 56.95±0.44 |
> | HPL (Semantic) | 72.86±1.89 | 74.13±1.88 | **60.74**±1.08 | 30.00±1.00 | **60.39**±0.74 | **58.50**±1.00 | **59.44**±0.63 |
> | **Qwen2.5-7B-Instruct** | 38.57 | 45.52 | 56.61 | 19.50 | 8.80 | 8.50 | 29.58 |
> | SFT | 67.62±2.18 | 73.63±3.11 | 60.64±1.12 | 31.83±1.26 | 66.70±1.11 | 65.17±0.76 | 60.93±0.71 |
> | RFT | 71.43±1.89 | 72.63±3.02 | 61.16±0.85 | 33.50±1.00 | 68.01±0.89 | 66.33±0.76 | 62.18±0.11 |
> | ETO | 72.62±2.51 | 77.86±2.40 | 61.85±1.00 | 33.17±1.04 | 68.32±0.86 | 67.00±0.50 | 63.47±0.47 |
> | IPR | 73.10±1.80 | 78.11±3.76 | 62.01±0.43 | 33.67±0.58 | 68.86±1.02 | 67.17±0.58 | 63.82±0.69 |
> | HPL (Fixed-N(3)) | 78.33±2.51 | 78.86±2.40 | 62.11±0.41 | 34.33±0.76 | 69.55±1.38 | 68.00±1.00 | 65.20±0.38 |
> | HPL (Fixed-K(3)) | **85.71**±2.58 | 78.61±1.55 | 62.01±1.04 | 33.83±1.26 | 69.40±0.98 | 68.17±0.58 | 66.29±0.54 |
> | HPL (Uncertainty) | 83.10±1.80 | 83.33±1.88 | 62.79±0.85 | **35.33**±1.04 | 69.21±0.47 | 67.83±0.29 | 66.93±0.43 |
> | HPL (Semantic) | 82.62±2.30 | **84.08**±2.28 | **62.97**±0.50 | 35.17±0.58 | **70.37**±1.27 | **68.50**±1.32 | **67.28**±0.47 |
>
> > **[W3] The proposed method seems to be much more complicated than the compared baselines, but the potential limitation coming from this approach is not well analyzed. HPL requires expert trajectory, generations from the reference model, reward signal, and access to a powerful model such as gpt-4o when semantic segmentation is used.**
>
> We appreciate the opportunity to clarify this practical aspect of HPL.
>
> - **Data requirements**: HPL uses the same expert trajectories as ETO and IPR. It **does not require additional expert data**; it simply reorganizes the existing trajectories into group-level units and constructs preference pairs at multiple granularities.
>
> - **Reference model generations & reward signals**: In fact, both ETO and IPR rely on a fixed reference policy to generate rollouts and obtain reward signals during the data generation phase (for generating negative data). This is not an additional requirement unique to HPL.
>
> - **Semantic segmentation**: Semantic segmentation with GPT-4o is an optional, best-performance variant. The simpler Fixed-N, Fixed-K and Uncertainty segmenters **do not rely on any external powerful LLM** and still yield gains over ETO and IPR. We will make this distinction clearer and add a table (as you suggested) comparing SFT / ETO / IPR / HPL variants in terms of external LLM usage, number of new generations, and wall-clock training time.

---

> ### Author Response · Authors · 2025-11-19
> **Response to Reviewer o3bh (2/3)**
>
> > **[W4] Some of the crucial experimental details seem to be omitted from the paper (ex. the value of $\gamma$ for the experiments, the number of generations $M$ used to evaluate the reward in Equation 2).**
>
> We apologize for omitting these details and will include them in the revised paper.
>
> - **Discount factor $\gamma$**: Prop. 1 is derived under a generic discounted MDP with $\gamma\in[0,1)$. In our experiments, the environment provides a sparse final outcome reward: a non-zero reward is given only at the final step, and all preceding steps yield zero reward. This setting can in fact be interpreted as a special case in which $\gamma$ approaches 1 over a finite horizon, since **it is equivalent to modeling the process with a discount factor $\gamma$ close to 1 together with appropriately scaled rewards**. The analysis remains conceptually valid because it relies solely on the existence of a bounded discounted return. In the revised version, we explicitly clarify the relationship between the $\gamma$ used in the theoretical analysis and the undiscounted experimental setting in Section 4.1.
>
> - **Number of MC rollouts $M$**: In all experiments we use $M=5$ rollouts per group. We report it in Section 4.1 in the revised version.
>
> We appreciate your attention to these details and will ensure they are fully documented.
>
> > **[Q1] How many additional datapoints were created by action group-level data generation in Section 3.2.2? How big is the dataset compared to step-level and trajectory-level datasets?**
>
> Thank you for the question. Below we report the dataset sizes for trajectory-level, step-level, and group-level preference data, measured on a single run with Qwen2.5-1.5B-Instruct for each benchmark.
>
> **Table 2.** Dataset sizes for trajectory-, step-, and group-level preference data (Qwen2.5-1.5B-Instruct, single run, Semantic segmenter).
>
> | Benchmark | Trajectory-level datapoints ($\mathcal{D}_\text{traj}$) | Step-level datapoints ($\mathcal{D}_\text{step}$) | Group-level datapoints ($\mathcal{D}_\text{group}$) |
> |---|---|---|---|
> | ALFWorld | 1,166 | 4,167 | 4,029 |
> | WebShop  | 606 | 1,312 | 1,058 |
> | InterCode-SQL | 805 | 1,665 | 1,259 |
>
> > **[Q2] If the value of $\gamma$ actually used for experiment was practically 1 (no discount in the trajectories), does the analysis of Proposition 1 not become incoherent?**
>
> As noted above ([W4]), Prop. 1 is stated for a generic discounted MDP with $\gamma\in[0,1)$ and bounded rewards. In practice, our environments provide undiscounted final rewards, which correspond to a finite-horizon episodic MDP; **this can be equivalently modeled with a discount $\gamma$ close to 1 and appropriately rescaled rewards**. The key assumptions required by Prop. 1 (bounded returns and a finite horizon) are satisfied in our setting. We will clarify this modeling choice and explicitly connect the theoretical $\gamma$ in Section 4.1.

---

> ### Author Response · Authors · 2025-11-19
> **Response to Reviewer o3bh (3/3)**
>
> > **[Q3] How much money in total was spent for the API call for semantic segmentation of the action groups, per each train-test run in Table 1?**
>
> Thank you for raising this practical question. Semantic segmentation is applied **only once per expert trajectory**, and each call to GPT-4o processes the full text transcript of a trajectory. In our experiments, the total number of expert trajectories is on the order of a few thousand per benchmark, so the overall API cost of semantic segmentation is modest compared to the GPU time required for fine-tuning 1.5B/7B models.
>
> We provide a cost estimate (based on token counts and current GPT-4o pricing) for each benchmark, and we emphasize that HPL can also operate with **purely heuristic segmenters** (Fixed-N, Fixed-K, and Uncertainty) that **incur no such API cost** while still delivering substantial improvements over the baselines.
>
> **Table 3.** Estimated GPT-4o API cost for semantic segmentation per benchmark.
>
> | Benchmark | # Expert Trajectories | Total Tokens | Estimated Cost (USD) |
> |:---:|:---:|:---:|:---:|
> | ALFWorld | 3,020 | ~2,183,735 | ~$6.55 |
> | WebShop | 1,824 | ~1,959,036 | ~$5.88 |
> | InterCode-SQL | 1,481 | ~1,009,963 | ~$3.03 |
>
> *(Current GPT-4o (Standard) pricing: 2.50 USD per 1M input tokens; 10.00 USD per 1M output tokens.)*
>
> > **[S1] Could you provide a table which present a comparison between HPL and the presented baselines, in terms of the use of external powerful LLM, number of new generations for the training, and wall-clock time for generation and training phase?**
>
> We appreciate your valuable suggestion and will incorporate such a table into Appendix C.5. We also present it here for your reference.
> The table, using Qwen2.5-1.5B-Instruct on the ALFWorld benchmark as an example, summarizes for each variant of SFT, ETO, IPR, and HPL:
> (i) whether a strong external LLM is used and for what purpose;
> (ii) the approximate number of LLM calls required in the data generation stage; and
> (iii) the approximate time needed for data generation and training.
> This will provide readers with a clearer understanding of the computational trade-offs.
>
> **Table 4.** Resource comparison of SFT, ETO, IPR, and HPL variants on ALFWorld benchmark with Qwen2.5-1.5B-Instruct.
>
> | Method | External powerful LLM | # LLM calls | Gen time | Train time |
> |---|---|---|---|---|
> | SFT | ✗ (uses logged expert demos only) | 0 | 0 | 18min |
> | ETO | ✗ (uses logged expert demos only) | ~30,000 | 1h 7min | 13min |
> | IPR | ✗ (uses logged expert demos only) | ~750,000 (step-level part) | 6h 13min (step-level part) | 26min |
> | **HPL (Fixed-N(3))** | ✗ (uses logged expert demos only) | ~207,000 (group-level part) | 3h 35min (group-level part) | 25min |
> | **HPL (Fixed-K(3))** | ✗ (uses logged expert demos only) | ~213,000 (group-level part) | 4h 15min (group-level part) | 26min |
> | **HPL (Uncertainty)** | ✗ (uses logged expert demos only) | ~194,000 (group-level part) | 3h 47min (group-level part) | 28min |
> | **HPL (Semantic)** | ✓ | ~221,000 (group-level part) | 3h 21min (group-level part) | 26min |
>
> *(Note: During data generation, we adopt the same parallel sampling implementation provided by the ETO and IPR codebases to accelerate environment interaction. The actual generation time depends primarily on the degree of parallelism in environment rollouts, as well as the time required to reset the environment and progress it to the desired states; the LLM call latency is not the dominant factor. The reported training times are based on real runs using 4 NVIDIA A800 80G GPUs.)*
>
> Thank you once more for your time and thoughtful review. We hope that our responses and new results resolve your main concerns.

---

### Official Review · Reviewer_8Yvn · 2025-11-10

**Soundness:** 2
**Presentation:** 2
**Contribution:** 2
**Rating:** 6
**Confidence:** 2

**Summary:**

This paper proposes Hierarchical Preference Learning (HPL) to address the granularity mismatch in preference-based offline alignment for long-horizon LLM agents. Conventional Direct Preference Optimization (DPO) methods operate either at the trajectory level or the step level. HPL introduces an intermediate group-level DPO that decomposes trajectories into semantically coherent action groups, paired with a dual-layer curriculum along task complexity and sample difficulty. Experiments on ALFWorld, WebShop, and InterCode-SQL demonstrate consistent improvements over existing benchmarks.

**Strengths:**

1. This work is well motivated. The paper starts with an intuitive problem, the granularity mismatch in preference optimization, and then proposes a novel solution that bridges trajectory/step-level supervisions.

2. The sample code is provided. It facilitates the reviewer to verify the numerical results independently.

3. The numerical performance is sound. The proposed methods beat the existing baselines by a clear margin.

**Weaknesses:**

1. The presentation could be improved.  While technically rich, some methodological descriptions (e.g., the Monte Carlo approach in reward estimation and curriculum scheduler) could benefit from more intuitive explanations.

**Questions:**

Please see the weakness part.

At the current stage, I tend to recommend acceptance. However, I'm open to revising my final decision after the rebuttal and further discussions.

---

> ### Author Response · Authors · 2025-11-19
> **Response to Reviewer 8Yvn**
>
> Many thanks for your positive comments and constructive feedback. Below we address each comment in detail and hope our responses help clarify and resolve your concerns.
>
> > **[W1] The presentation could be improved. While technically rich, some methodological descriptions (e.g., the Monte Carlo approach in reward estimation and curriculum scheduler) could benefit from more intuitive explanations.**
>
> Thank you for your suggestion. We agree that some methodological components, particularly the MC reward estimation and the dual-layer curriculum, could benefit from more intuitive exposition.
>
> In the revision, we:
> - Add a short **illustrative example** in **Appendix D** that walks through how MC rollouts from a given group $G_i$ are used to estimate its expected final outcome reward, using a concrete ALFWorld trajectory.
> - Provide pseudocode for the dual-layer curriculum scheduler in **Appendix E** showing how groups move through the 2D length–difficulty matrix across the three training phases.
>
> We believe these changes will make the paper more accessible to readers.
>
> Thank you once more for your time and thoughtful review. We hope that our responses address your main concerns.

---

### Author Response · Authors · 2025-12-01
**Overall response to reviews and revision summary**

Dear Area Chair and Reviewers,

We sincerely thank the Area Chair and all reviewers for their time and constructive comments. During the discussion period, we studied all reviews, provided point-by-point responses, conducted additional experiments (new baselines, robustness checks), and revised the manuscript accordingly.

As the reviews highlighted, our work is seen as **tackling the granularity mismatch in preference-based alignment for long-horizon LLM agents**, and HPL as a **hierarchical framework** combining trajectory-, step-, and group-level preferences with curriculum learning. Reviewers found the paper **clear and well organized**, with **informative ablations**, and noted that HPL offers **principled mid-granularity learning with a sound bias–variance trade-off** and **consistent improvements over baselines on the benchmarks**, with **released code** for verification.

We thank the reviewers for highlighting the need for clearer **motivation and framing of the trajectory–step trade-off** (Reviewer 9rwf, Reviewer Q8y5), a more precise **problem setting and terminology for one-shot data collection vs. online RL** and **more complete baselines including online RL / IRL** (Reviewer 9rwf), **robustness and sensitivity analysis for segmentation and curriculum** (Reviewer Q8y5, Reviewer o3bh, Reviewer 8Yvn), **reporting standard deviations, dataset scales, and key hyperparameters** (Reviewer o3bh), **clarifying computational cost and external LLM usage** (Reviewer Q8y5, Reviewer o3bh, Reviewer 8Yvn), and **more intuitive explanations of Monte Carlo reward estimation and the curriculum** (Reviewer 8Yvn). In response, we provided detailed clarifications in our rebuttal and revised the manuscript with the following updates:

- **[Reviewer 9rwf] Motivation and trajectory–step trade-off.**
  We rewrote the motivation around the **granularity trade-off** between trajectory- and step-level DPO, removed "myopic/local feedback" wording, and framed limitations in realistic data- and rollout-limited settings with segmentation and group-level objectives as key inductive biases.

- **[Reviewer 9rwf] Problem setting and terminology.**
  We added a **Problem Setting** subsection formalizing a two-stage protocol (one-shot exploration + offline preference optimization) and replaced "offline RL" with **preference-based methods with one-shot data collection**, clearly separating this from fully online PPO/GRPO.

- **[Reviewer 9rwf, Reviewer Q8y5, Reviewer o3bh] Theory and finite-sample view.**
  We clarified the assumptions of Proposition 1 and refined the discussion of step-level Monte Carlo estimates to emphasize a **limited rollout budget** and the variance / credit-assignment benefits of group-level aggregation.

- **[Reviewer 9rwf] Baselines: online RL and IRL.**
  We added **PPO, GRPO**, and an **IRL-style baseline (InversePRM)** under comparable data budgets, showing that HPL remains stronger while operating in the one-shot data collection regime.

- **[Reviewer Q8y5] Robustness to segmentation and curriculum.**
  We showed that four segmentation strategies (Fixed-N, Fixed-K, Uncertainty, Semantic) all improve over baselines and added a **curriculum sensitivity study** (±50% thresholds) demonstrating stable gains over the non-curriculum variant.

- **[Reviewer o3bh] Standard deviations, dataset scale, and hyperparameters.**
  We now report **mean ± standard deviation** over three seeds, add dataset size statistics for trajectory-, step-, and group-level preferences, and list key hyperparameters (*e.g.*, number of Monte Carlo rollouts).

- **[Reviewer Q8y5, Reviewer o3bh] Computational cost and external LLM usage.**
  We added a **resource comparison table** and a **GPT-4o cost estimate** for semantic segmentation, and emphasized that heuristic segmenters (Fixed-N/K, Uncertainty) use **no external LLM** while capturing most of HPL's gains.

- **[Reviewer 8Yvn, Reviewer 9rwf] Presentation and discussion.**
  We improved explanations of Monte Carlo reward estimation and the curriculum with concise examples and pseudocode, and added a discussion section emphasizing **segmentation as structural prior** and **statistical properties of the group-level objective**.

During the discussion phase, **Reviewer Q8y5** noted that our rebuttal **"really helped clarify things"**, that the hierarchical aspect **"finally clicked"**, and that the extra ablations and cost breakdown were **"actually pretty convincing"**, and accordingly **raised the score from 4 to 6**. **Reviewer 9rwf** reported that the concerns were **"partially resolved"** and gave concrete guidance on revising the motivation and problem setting. We fully adopted these suggestions and substantially revised the whole manuscript.

We hope that our responses, additional experiments, and revisions adequately address the reviewers' concerns and clarify the contribution and scope of HPL.

Many thanks,
Submission 5336 Authors

---

### Meta-Review · Area_Chair_8AUj · 2026-01-09

**Summary:**

The paper presents a novel approach called Hierarchical Preference Learning (HPL) aimed at addressing the granularity mismatch in preference-based offline alignment for long-horizon LLM agents. This approach introduces an intermediate group-level Direct Preference Optimization (DPO) that breaks down trajectories into semantically coherent action groups, thereby enhancing the efficiency and effectiveness of learning algorithms in complex environments.

Here are some strengths of the paper:

+ HPL's incorporation of a group-level DPO represents a significant advancement over traditional DPO methods that function at either the trajectory or step level. This mid-granularity approach is well-founded theoretically, as it provides a favorable bias–variance trade-off, achieving lower variance compared to conventional methods while maintaining a controlled bias.

+ The experiments conducted on diverse platforms—ALFWorld, WebShop, and InterCode-SQL—demonstrate consistent and significant performance improvements over existing benchmarks. The reported results, particularly the high performance in the ALFWorld-unseen scenario (86.57%), are compelling and validate the efficacy of the proposed method.

+ The paper is well-organized and clearly written, facilitating comprehension of the complex methodologies introduced. Effective visualizations, including Figure 2 which illustrates the HPL framework and training procedure, enhance the reader’s understanding of the proposed approach.

+ The inclusion of ablation studies, particularly illustrated in Figure 5, provides valuable insights into the contributions of various components of the HPL algorithm. This transparency in methodology strengthens the claims regarding the effectiveness of group-level loss and the dual-layer curriculum in improving learning outcomes.


However, the reviewers also raised some concerns:

- While the experiments show promising results across the selected environments, further evaluation in more diverse and real-world scenarios would strengthen the claims of generalizability and robustness of HPL.

- The introduction of a dual-layer curriculum based on task complexity and sample difficulty adds layers of complexity to the implementation of the algorithm. A more detailed discussion on the practical difficulties faced during implementation and potential solutions would be beneficial for practitioners.


The scores of the reviewers are 6,4,4,2, and one 4 is increased to 6 in the rebuttal. I belive most concerns have been alliviated by the authors. Thus, I recommend accept.

**Reviewer Concerns:**

Most were addressed.

**Reviewer Scores:**

Will change.

---

### Decision · Program_Chairs · 2026-01-26

Accept (Poster)